# The impact of rearing environment on *C. elegans*: phenotypic, transcriptomic and intergenerational responses to 3D enriched habitats

Aurélie Guisnet[1], Nour Halaby[1], Maxime Rivest[1], Beatriz Romero Quineche[2] and Michael Hendricks[1],*

## ABSTRACT

Environmental context profoundly influences organismal biology, yet laboratory studies often rely on simplified conditions that may not fully capture natural phenotypic repertoire. This exploratory study investigated how rearing environment affects various aspects of *Caenorhabditis elegans* biology by comparing worms cultured in three-dimensional decellularized fruit tissue scaffolds with those raised on standard two-dimensional agar plates. While fat content and feeding rate remained stable across conditions, other life history traits demonstrated varying degrees of plasticity in response to environmental context. We observed that scaffold-grown worms exhibited reduced body size, altered reproductive strategies, and mild enhancements in stress resistance, burrowing ability, swimming kinematics and exploratory behavior. RNA sequencing revealed distinct transcriptional profiles between scaffold-grown and agar-grown worms, with most changes arising within one generation. Some traits showed evidence of intergenerational inheritance. Our findings highlight the sensitivity of *C. elegans* biology to rearing conditions and underscore the importance of considering environmental context in interpreting laboratory results. This work sets the foundation for future research into the mechanisms underlying environmental adaptation and phenotypic plasticity in model organisms.

**KEY WORDS: *Caenorhabditis elegans*, Plasticity, Environmental enrichment, Rearing environment, RNA sequencing, Intergenerational effects**

## INTRODUCTION

Environmental conditions are fundamental drivers of biological development, behavior, and physiology across the animal kingdom. Organisms in the wild navigate complex, heterogeneous habitats that exert multifaceted influences on their phenotypes (Renner and Rosenzweig, 1987). The functional role of many genes depends critically on environmental context: for example, 80% of the yeast genome can be deleted without causing inviability under optimal laboratory conditions, yet under environmental stress, this fraction drops to only 3% (Hillenmeyer et al., 2008). However, laboratory studies often rely on simplified, uniform conditions to facilitate control and reproducibility, potentially limiting our understanding of how environmental variability shapes biological processes and raising important questions about the breadth and applicability of our findings (Petersen et al., 2015; Würbel, 2000). This discrepancy is particularly evident in the study of model organisms like *Caenorhabditis elegans*, which are conventionally cultured on two-dimensional (2D) agar plates with uniform bacterial lawns – a stark contrast to their wild, dynamic, three-dimensional (3D) environments rich in structural and microbial diversity (Brenner, 1974; Schulenburg and Félix, 2017).

Previous research has underscored that minimalistic laboratory conditions can significantly affect gene expression patterns and behaviors, potentially obscuring the full spectrum of an organism's biological responses (Alfred and Baldwin, 2015; Voelkl et al., 2020). Recognizing this limitation, there is a growing interest in exploring how varying environmental contexts influence organismal biology (Petersen et al., 2015). In mammals, fish, and even insects, environmental enrichment, the addition of structural complexity, sensory stimulation, or opportunities for exploration, can enhance learning, stress resistance, and neuroplasticity while reducing stereotypic behaviors (Mallory et al., 2016; Mason et al., 2007; Nithianantharajah and Hannan, 2006; Salvanes et al., 2013). Whether similar effects occur in organisms with simpler nervous systems remains an open question. Altering the rearing environment, even within controlled laboratory settings, can provide insights into the adaptability and plasticity of biological systems, shedding light on gene–environment interactions that are otherwise overlooked (Renner and Rosenzweig, 1987). In our prior work, we introduced a novel method for cultivating *C. elegans* within 3D decellularized fruit tissue scaffolds, aiming to create a more complex, enriched living environment while maintaining experimental control (Guisnet et al., 2021a). This approach allowed us to observe behaviors and phenotypes that differed from those exhibited under standard laboratory conditions, suggesting that environmental context plays a significant role in shaping biological outcomes. Specifically, worms displayed a preference for the enriched 3D environment and demonstrated complex dauer behaviors not commonly observed in conventional settings. Despite these initial findings, the broader implications of raising *C. elegans* in alternative environments remain largely unexplored. How does exposure to an enriched habitat influence various aspects of their biology, such as feeding behavior, reproduction, development, stress resistance, locomotion, and gene expression? Moreover, can these environmentally induced traits persist across generations, indicating potential mechanisms of intergenerational inheritance?

The present exploratory study aims to address these questions by examining the effects of the rearing environment on a suite of phenotypic and behavioral traits in *C. elegans*. By comparing worms cultured in our enriched scaffolds with those raised under

[1]McGill University, Department of Biology, Montréal, H3A 1B1, Canada. [2]University of Ottawa, 75 Laurier Av. E, Ottawa, ON, K1N 6N5, Canada.

*Author for correspondence (michael.hendricks@mcgill.ca)

A.G., 0000-0002-1102-6017; M.R., 0000-0002-1196-4679; M.H., 0000-0002-3408-3858

Biology Open

standard laboratory conditions, we seek to explore how environmental context influences biological processes. Additionally, we investigate whether the observed phenotypic changes are heritable, contributing to the understanding of how environmental factors can lead to intergenerational phenotypic variations (Rechavi and Lev, 2017). Based on findings from other taxa and stress studies in *C. elegans*, we anticipated that behavioral traits would show greater plasticity than core physiological parameters (Ganguly-Fitzgerald et al., 2006; Kwon et al., 2015; Laranjeiro et al., 2019; Mason and Rushen, 2006; Rampon et al., 2000; van Praag et al., 2000; Vidal-Gadea et al., 2012; Volgin et al., 2018). We further expected that most environmentally induced changes would arise within a single generation as *C. elegans* have evolved to thrive in rapidly changing environments, though some traits might show evidence of intergenerational inheritance as it has been previously shown with inheritance of stress responses (Cheung et al., 2005; Fire et al., 1998; Frézal and Félix, 2015; Posner et al., 2019; Swierczek et al., 2011; van der Linden et al., 2010).

By providing an exploratory characterization of the influence of the rearing environment on *C. elegans* biology, this study aims to lay the groundwork for future research into the mechanisms underlying these effects. Using a compound phenotypic divergence analysis of ten biological modules across life history, behavior and genetics, we find that the immediate rearing environment exerts a significant global effect on phenotype, while ancestral environment shows a modest, non-significant influence. These factors act largely independently, with minimal interaction effects. Individual traits show varying levels of stability and responsiveness to environmental context. Understanding the full spectrum of *C. elegans* biological responses to different environmental contexts is crucial not only for interpreting laboratory findings more accurately but also for leveraging this model organism to study fundamental questions about gene-environment interactions, phenotypic plasticity, and the evolution of adaptive traits. This work highlights the importance of incorporating environmental context into the interpretation of biological research to achieve a more comprehensive understanding of life sciences.

## RESULTS

To ensure that phenotypic effects were not confounded by residual toxicity from the fruit tissue preparation process (Guisnet et al., 2021b), we analyzed the composition of the decellularized scaffold matrix for traces of the toxic decellularizing agent, sodium dodecyl sulfate (SDS), using high resolution magic angle spinning nuclear magnetic resonance spectroscopy (HR-MAS). We found no traces of SDS in the scaffold (Fig. S1), indicating that animals are not affected by detergent residues. Peaks identified were consistent with components of plant cell walls or their breakdown products.

To disentangle the immediate effects of the growing environment from potential intergenerationally inherited traits, we compared four groups throughout: worms with an agar ancestry and raised on agar (agar:agar); worms with an agar ancestry, but raised on scaffold (agar:scaffold); worms with a scaffold ancestry, but raised on agar (scaffold:agar); and worms with a scaffold ancestry and raised on scaffold (scaffold:scaffold) (Fig. 1A). Ancestry was established by raising the parents of all tested animals for at least ten consecutive generations in their respective scaffold or conventional agar habitat. This design allowed us to assess both the direct impact of the habitat and any epigenetic influences transmitted across generations.

### Compound phenotypic analysis reveals rearing environment as the dominant driver of divergence

Given the exploratory nature of this study and the large number of phenotypic measurements across diverse biological traits, we sought to first establish whether the immediate growth environment and ancestry have detectable effects on global *C. elegans* biology. Rather than conducting multiple hypothesis tests across measurements (see Materials and Methods), we computed a Phenotypic Divergence Statistic ($D^2$) that integrates effect sizes across all measurements into a single omnibus test (Fig. 1B). We defined ten biological modules across life history, behavior and genetics, encompassing feeding rate, body morphology, fat content, reproductive output, egg morphology, burrowing ability, oxidative stress resistance, swimming behavior, crawling behavior, and transcriptomic profiles. For each module, we extracted $t^2$-equivalent effect sizes for the rearing environment, ancestry, and their interaction to obtain global $D^2$ values. Permutation testing revealed that the immediate growth environment had a substantial and statistically significant effect on overall worm phenotype ($D^2_{growth}=76.43$, $P=0.001$), while ancestry showed a more modest effect that did not reach significance ($D^2_{ancestry}=13.39$, $P=0.221$). The interaction between rearing environment and ancestry was minimal ($D^2_{interaction}=7.35$, $P=0.608$), suggesting that these factors act largely independently. Having established these global patterns, we next detail each biological measurement individually to explore traits contribution and guide future research.

### Pumping rate is similar across environmental context

The pumping rates across all conditions showed a relatively narrow range, with most values falling between 40 and 50 pumps per 10 s (Fig. 2A).

### Morphological development is altered by experience with the enriched environment

To characterize the effects on development, we imaged worms every 24 h after egg laying over 6 days and measured their morphological features.

Scaffold-grown worms, regardless of ancestry, showed reduced body length starting at day 3 compared to agar-grown worms, while lengths on days 1 and 2 were more similar (Fig. 2B). We also noted a steeper increase in length for agar-grown animals starting at day 4 compared to scaffold-grown animals, which showed a milder increase (Fig. 2B). Mean body width revealed a similar pattern, with scaffold-grown worms generally being narrower than their agar-grown counterparts, although throughout the 6 days of development, and we observed for all groups a slight decrease in width after day 4 (Fig. S2A). Mid-body width showed similar trends as mean body width (Fig. S2B), and worm area and perimeter similar to body length (Fig. S2C,D).

To assess overall body proportions, we calculated the ratio of body length to mean width for each animal. Despite the differences in absolute size described above, the body proportions of scaffold-grown and agar-grown worms remained similar on average (Fig. 2C), but scaffold-grown worms tended to be thinner in their early life and showed more variability in body proportions during early development. However, body proportions were more similar later in development (Fig. 2D). Scaffold:agar worms had a strong tendency to be wider than agar:agar worms on day 2, but had similar proportions on other days (Fig. 2D). The increased variability in scaffold-grown worms was also evidenced by the size differences in the early days of life, which could be less easily distinguished than in agar-grown animals (Fig. S2E).

### Fat content is not affected by environmental context

To explore impacts on metabolism, we performed Oil Red O staining to assess major fat stores in worms of mixed life stages (O'Rourke et al., 2009).

We analyzed the staining intensity of individual worms relative to their body size. While the highest intensities were observed in agar-

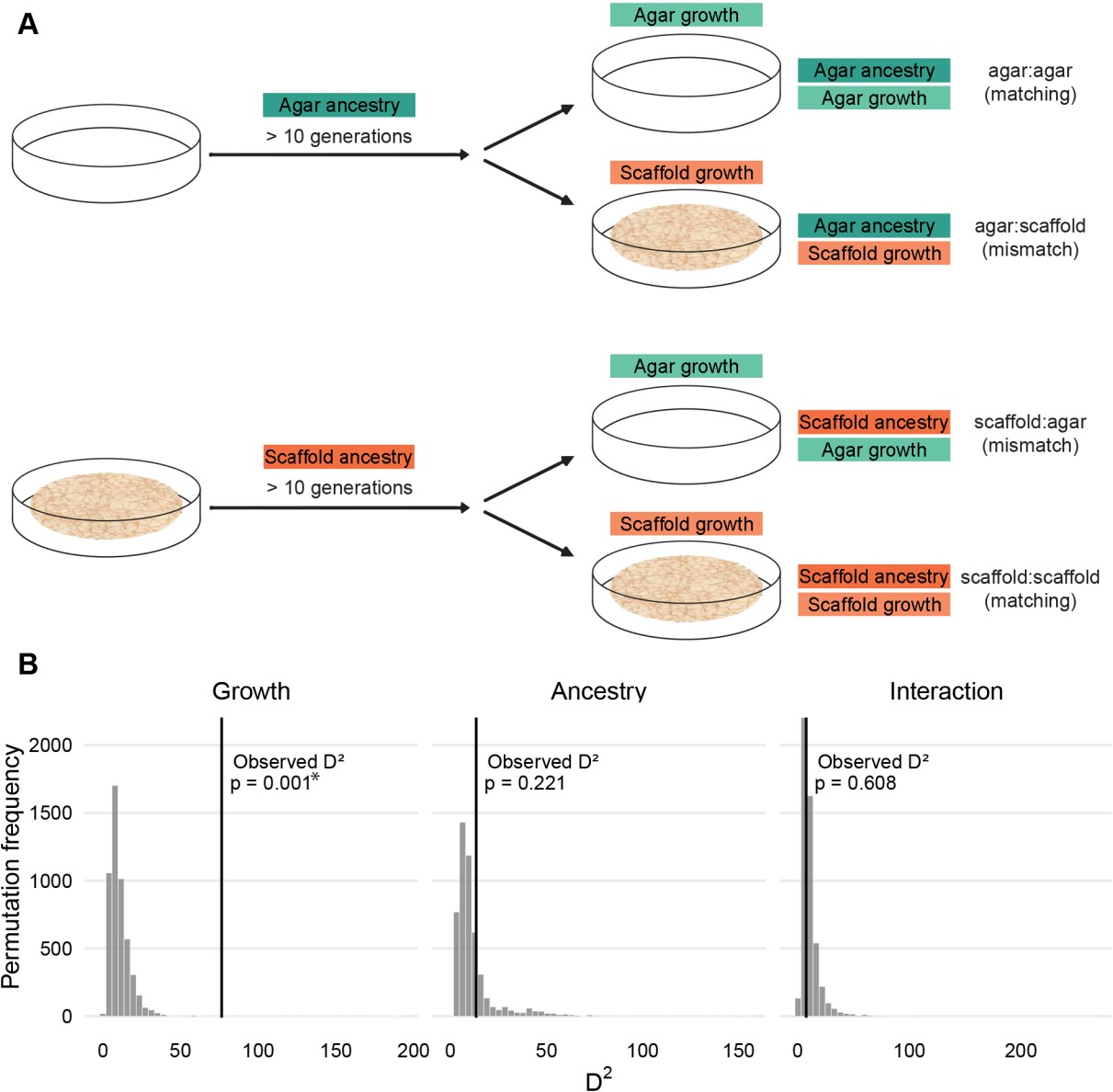

**Fig. 1. Experimental design investigating the effects of rearing environment and ancestry.** (A) Ancestry was established by raising the parents of all tested animals for at least ten consecutive generations in their respective scaffold or conventional agar habitat. The growing habitat of experimental worms was established by moving gravid adults to either a matching or mismatched habitat to lay their eggs, leading to four experimental conditions: worms with an agar ancestry and raised on agar (agar:agar), worms with an agar ancestry, but raised on scaffold (agar:scaffold), worms with a scaffold ancestry, but raised on agar (scaffold:agar), and worms with a scaffold ancestry and raised on scaffold (scaffold:scaffold). (B) Histograms show the null distributions of the compound Phenotypic Divergence Statistic ($D^2$) generated from 5000 permutations, where factor labels were shuffled within each biological module and models were refit. The x-axis represents $D^2$ values where higher values indicate greater overall phenotypic divergence. Black vertical lines indicate the observed $D^2$ values from the actual data. Permutation $P$-values were calculated as the proportion of permuted $D^2$ values equal to or greater than the observed value. Rearing environment showed a significant global effect on phenotype ($D^2$=76.43, $P$=0.001), while ancestry ($D^2$=13.39, $P$=0.221) and the rearing×ancestry interaction ($D^2$=7.35, $P$=0.608) did not reach significance. *=$P$<0.05.

grown worms only, our results suggest that staining intensity was dependent on worm size across all conditions, where larger worms generally showed higher staining intensities (Fig. 2E).

### Environmental context and ancestry modulate total brood size and temporal patterns

We next tested whether the environment had an effect on total brood size and the distribution of egg-laying over time. We found that scaffold:scaffold worms had a smaller total brood than traditional agar:agar worms (Fig. 3A). In addition, we found that both groups, once at the egg-laying stage, were unaffected by the egg-laying

habitat (Fig. 3B). The smaller brood size in scaffold:scaffold worms was mainly on the first and second days of egg-laying (Fig. 3C).

For worms raised in a habitat mismatched to their ancestors, both agar:scaffold and scaffold:agar worms produced a similar number of offspring as agar:agar worms (Fig. 3A).

### Egg production dynamics and morphology are affected the enriched habitat

To investigate further differences in development and brood size, we counted the number of eggs present in the gonads of individual worms over the 6 days after being laid. Eggs became visible in the

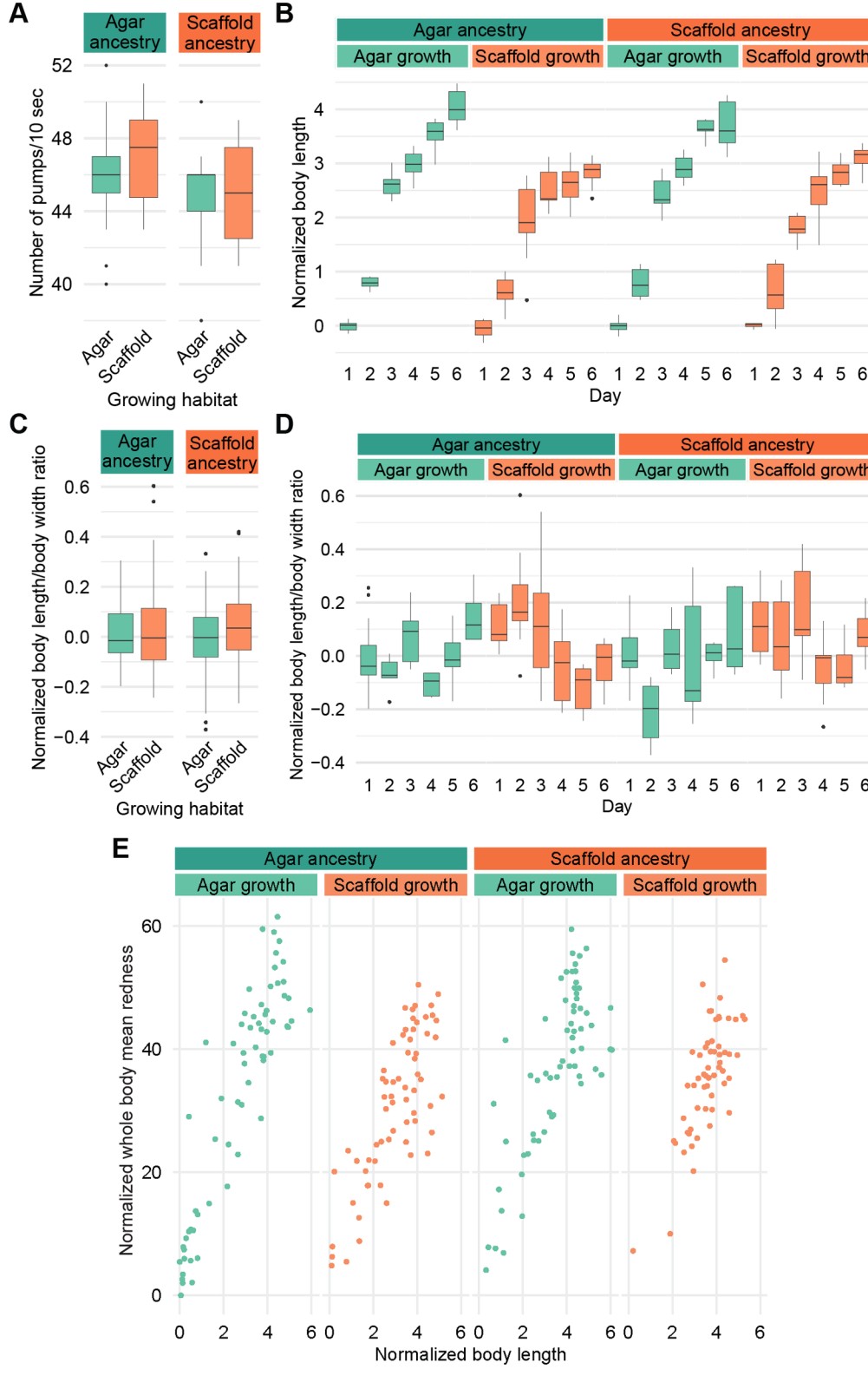

**Fig. 2. Environmental context modulates body size and developmental trajectories but not feeding rate or fat storage.**
(A) Number of pharyngeal pumps per 10 s. Sample size: agar:agar: $n$=18; agar:scaffold: $n$=16; scaffold:agar: $n$=19; scaffold:scaffold: $n$=19. (B) Body length by day. (C) Ratio of worm length by mean body width, all 6 days combined. (D) Ratio of worm length by mean body width, by day. Sample size for B–D: agar:agar: day 1: $n$=13, day 2: $n$=7, day 3: $n$=12, day 4: $n$=10, day 5: $n$=11, day 6: $n$=7; agar:scaffold: day 1: $n$=8, day 2: $n$=17, day 3: $n$=10, day 4: $n$=12, day 5: $n$=8, day 6: $n$=10; scaffold:agar: day 1: $n$=12, day 2: $n$=8, day 3: $n$=11, day 4: $n$=9, day 5: $n$=6, day 6: $n$=7; scaffold:scaffold: day 1: $n$=7, day 2: $n$=14, day 3: $n$=11, day 4: $n$=10, day 5: $n$=6, day 6: $n$=9. (E) Mean whole body redness by body length stained by Oil Red O. Sample size, one point per worm: agar:agar: $n$=66; agar: scaffold: $n$=65; scaffold:agar: $n$=64; scaffold:scaffold: $n$=57. All measured values in B–E are normalized to the respective mean of agar:agar worms on day 1.

gonads across all conditions only by day 3. The number of eggs present in the gonads were proportional to worm size, with worms accumulating more eggs in their gonads with size on days 3 and 4 (Fig. 3D). On days 5 and 6, when size was more stable (Fig. 2B), the number of eggs in their gonads was also stable across all conditions, despite scaffold-grown worms having smaller body sizes (Fig. 3D).

While total egg area was similar across all conditions, eggs from scaffold-grown worms showed greater elongation (Fig. 3E,F).

## Environmental context modulates burrowing behavior

To explore changes in neuromuscular performance and chemotaxis, we employed the burrowing assay described by Laranjeiro et al.

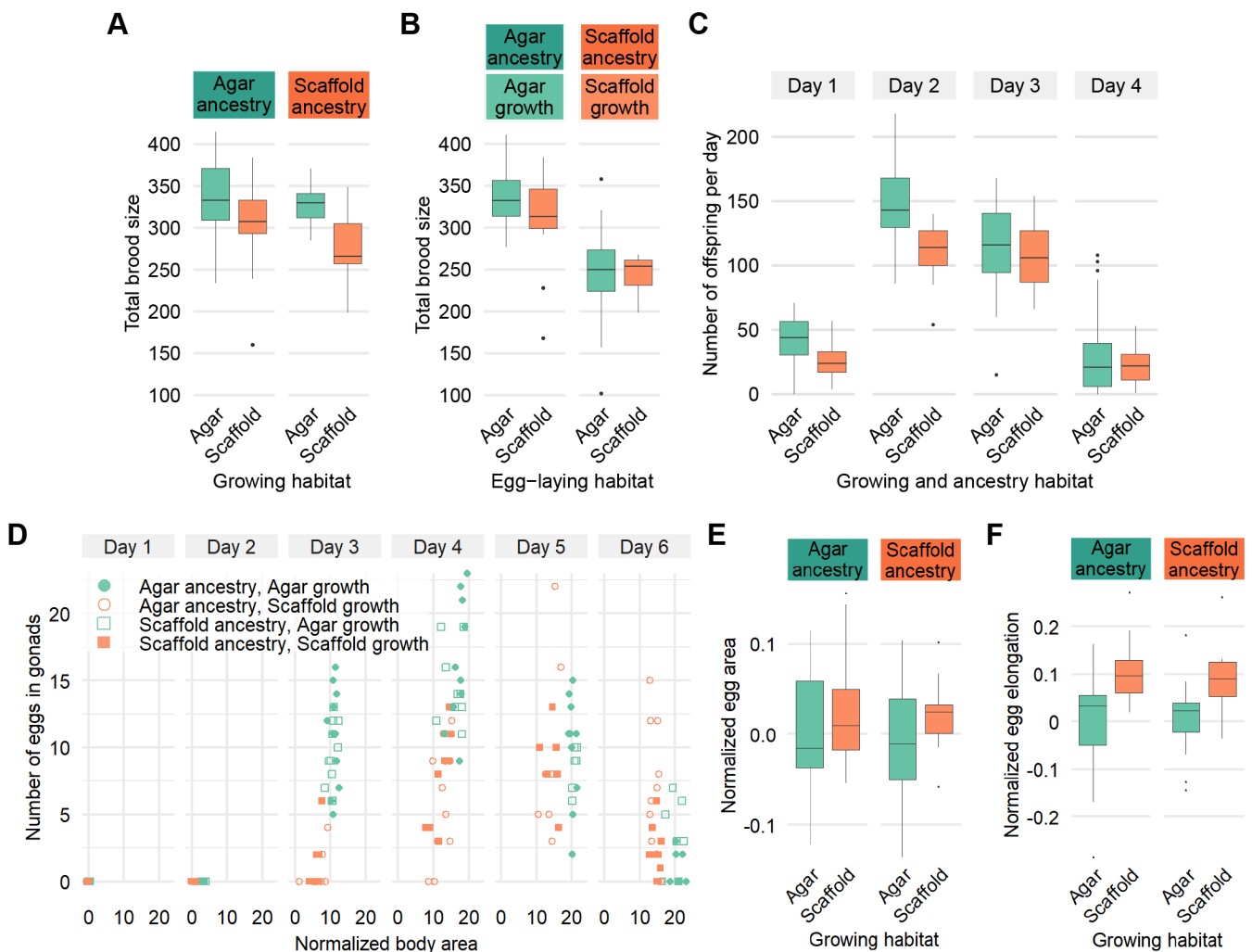

**Fig. 3. Rearing environment and ancestry influence reproductive strategies and egg morphology.** (A) Total brood size. Sample size: agar:agar: *n*=23; agar:scaffold: *n*=26; scaffold:agar: *n*=21; scaffold:scaffold: *n*=13. (B) Total brood size for agar:agar and scaffold:scaffold worms, but once at the egg-laying stage, they were moved to a habitat matching or mismatched of their own. Sample size: agar on agar: *n*=20; agar on scaffold: *n*=14; scaffold on agar: *n*=17; scaffold on scaffold: *n*=8. (C) Number of offspring per day. Sample size for each day: agar: *n*=43; scaffold: *n*=21. (D) Number of eggs in gonads by worm body area normalized to the mean of agar:agar worms on day 1, by day. Sample size: agar:agar: day 1: *n*=13, day 2: *n*=7, day 3: *n*=12, day 4: *n*=10, day 5: *n*=11, day 6: *n*=7; agar:scaffold: day 1: *n*=8, day 2: *n*=17, day 3: *n*=10, day 4: *n*=12, day 5: *n*=8, day 6: *n*=10; scaffold:agar: day 1: *n*=12, day 2: *n*=8, day 3: *n*=11, day 4: *n*=9, day 5: *n*=6, day 6: *n*=7; scaffold:scaffold: day 1: *n*=7, day 2: *n*=14, day 3: *n*=11, day 4: *n*=10, day 5: *n*=6, day 6: *n*=9. (B) Egg area, and (C) egg elongation, normalized to the mean of agar:agar worms on day 1. Sample size (B,C): agar:agar: *n*=14; agar:scaffold: *n*=13; scaffold:scaffold: *n*=12; scaffold:agar: *n*=13.

(2019) on young adult worms. This assay measures the ability of worms to navigate through a 3D pluronic gel matrix towards an attractant (*E. coli* OP50, in our case).

Of the animals that reached the attractant, their median time to reach the top had a broad, overlapping range. Scaffold:scaffold worms had a median time to top of only 40 min, while the other groups took 85–90 min (Fig. 4A). On the other hand, all conditions showed a similar total fraction of worms to have reached the attractant (between 67.9–73.3%) after 3 h, except for the agar: scaffold group, which had only 50% of the animals reach the top (Fig. 4B). Scaffold:scaffold worms particularly showed a larger percentage of worms reaching the top in the first 90 min (Fig. 4B).

### Scaffold-grown *C. elegans* exhibit a modest increase in oxidative stress resistance
To investigate the effects of the enriched habitat on stress resistance, we exposed young adult worms to oxidative stress using 3 mM

$H_2O_2$. Our results suggest a slight enhancement in oxidative stress resistance for worms grown on scaffold. Particularly, only scaffold: scaffold animals were still alive at the 6-h time point. No worm survived the entire 8-h period (Fig. 4C).

### Swimming behavior is modulated by ancestry and experience
We further looked into behavioral differences by examining swimming behavior of young adults. We used methyl cellulose (MC) to create a range of viscous liquids: 0% (1 cP, same as water), 0.5% (5.6 cP, similar to milk) and 1% (20 cP, similar to vegetable oil). Higher viscosities were not tested as above 20 cP, worms exhibited crawl-like motion.

We looked into long-term swimming patterns by counting swimming bends in 5 s periods over 3 h, as described in Ghosh and Emmons (2008). In the 0% MC solution, it was first evident that worms grown in a habitat mismatched from their ancestors spent

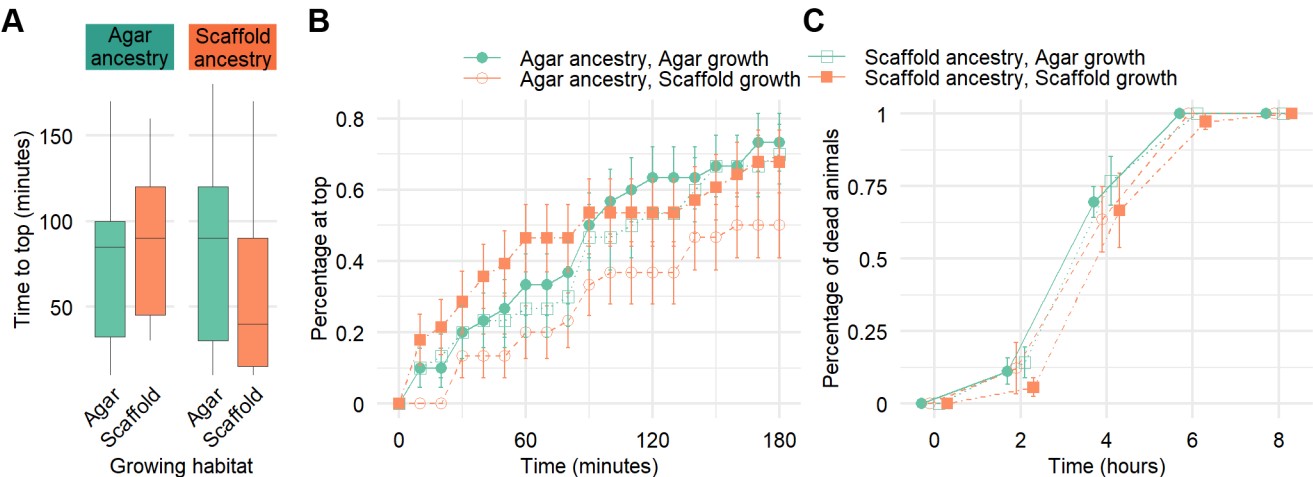

**Fig. 4. Burrowing ability and oxidative stress resistance are modestly modulated by environmental context and ancestry.** (A) Time to reach the attractant at the top of the gel, successful animals only. (B) Total percentage of worms having reached the attractant located at the top of the gel for every 10 min time-point over 180 min. Error bars represent standard error. Sample size (A,B): agar:agar: $n$=30; agar:scaffold: $n$=30; scaffold:scaffold: $n$=28; scaffold:agar: $n$=30. (C) Total percentage of dead worms for every 2 h time-point over 8 h. Error bars represent standard error. Sample size: agar:agar: $n$=36; agar:scaffold: $n$=35; scaffold:scaffold: $n$=36; scaffold:agar: $n$=36.

more time being quiescent (0–2 bends per 5 s) (Fig. 5A and Fig. S3A). There was no difference observed in the proportion of time spent in slow swimming (2–5 bends per 5 s) (Fig. 5B). However, of the periods where worms were swimming (5 bends or more per 5 s), scaffold-grown worms showed a slight increase in their average number of bends per 5 s compared to agar-grown worms, regardless of ancestry (Fig. 5C). A time series heatmap showed that worms grown in a habitat matching their ancestors had no quiescent periods in the last 30 min of the 3 h period (Fig. 5D). On the other hand, there was an observable increase in quiescence through time for animals grown in conditions mismatched from their ancestors (Fig. 5D). Still, all conditions showed slowing with time, although this trend was the strongest in scaffold:agar worms (Fig. S3B).

In the more viscous solutions, worms would often be immobile at the side of the well, perhaps due to the increased surface tension. Thus, when considering only swimming periods where worms were freely swimming, there was no marked difference in their speed in 0.5% MC, but a slight increase was observed in 1% MC again in scaffold-grown animals (Fig. 5C).

We next recorded young adult worms swimming in liquid droplets for 1 min in 0% MC, but for 30 s in 0.5% and 1% MC as they would rapidly settle on the edges of the droplet. We did not observe any difference in the proportion of time spent in different body shapes (O-shape, 6-shape, U-shape, C-shape or S-shape) (Fig. 5E and Fig. S3C) (Guisnet and Hendricks, 2025 preprint). Since scaffold-grown worms were on average smaller as adults than agar-grown worms, we calculated their bend amplitude as a proportion of their body length (Fig. S3D). This comparison revealed no marked difference between the conditions or viscosities (Fig. 5F). For frames where animals were in S-shape, no difference in body wavelength was observed either (Fig. 5G).

### Exploratory behavior is mildly affected by experience with the scaffold habitat

To investigate influences on locomotion on a 2D surface, we analyzed the crawling behavior of young adult worms for 1 min under three different food availability conditions: absence of food (none), low food density, and high food density.

Across all conditions, worms spent the majority of their time moving forward and spent slightly more time stationary than moving backward (Fig. 6A). However, the agar:agar worms in low food density, spent less time moving forward and more time stationary, resulting in nearly equal time allocation between these two states (Fig. 6A). The average speed during both forward and backward movement was not different between groups or food densities (Fig. 6B).

There was no difference in the proportion of time spent in different body shapes with S-shaped postures being overwhelmingly dominant and other shapes occurring infrequently and at similarly low levels (Fig. 6C). This suggests no difference in the frequency of deep turns which are characterized by O, U and 6-shapes (Guisnet and Hendricks, 2025 preprint). Similarly, analysis of head bend movements revealed no difference in the frequency of bends (Fig. S4A) or their depth (Fig. S4B).

On average, the total distance travelled during the recording period and the furthest distance reached did not differ considerably between the groups or conditions, except for scaffold:scaffold worms, which travelled approximately 30% further in the absence of food (Fig. 6D,E). While tortuosity was similar in the presence of food (low and high density), in the absence of food, worms raised in a habitat mismatched to their ancestors exhibited higher path tortuosity suggesting a prioritization of local search (Fig. 6F). In both low and high-density food environments, all groups reached their furthest exploratory point near the end of the recording period (Fig. S4C). In the absence of food, this trend remained true for scaffold:scaffold worms, but was slightly less pronounced for the other groups (Fig. S4C).

### Rearing environment drives mild, rapidly reversible changes transcriptional profiles

We investigated transcriptional changes by performing RNA sequencing on young adult worms. Principal component analysis revealed that rearing environment and ancestry were only mild drivers of gene expression differences, aligning with the subtle phenotypic changes observed in our study (Fig. 7A,B).

To explore the specific effects of the immediate growing environment and ancestry, we conducted factorial differential

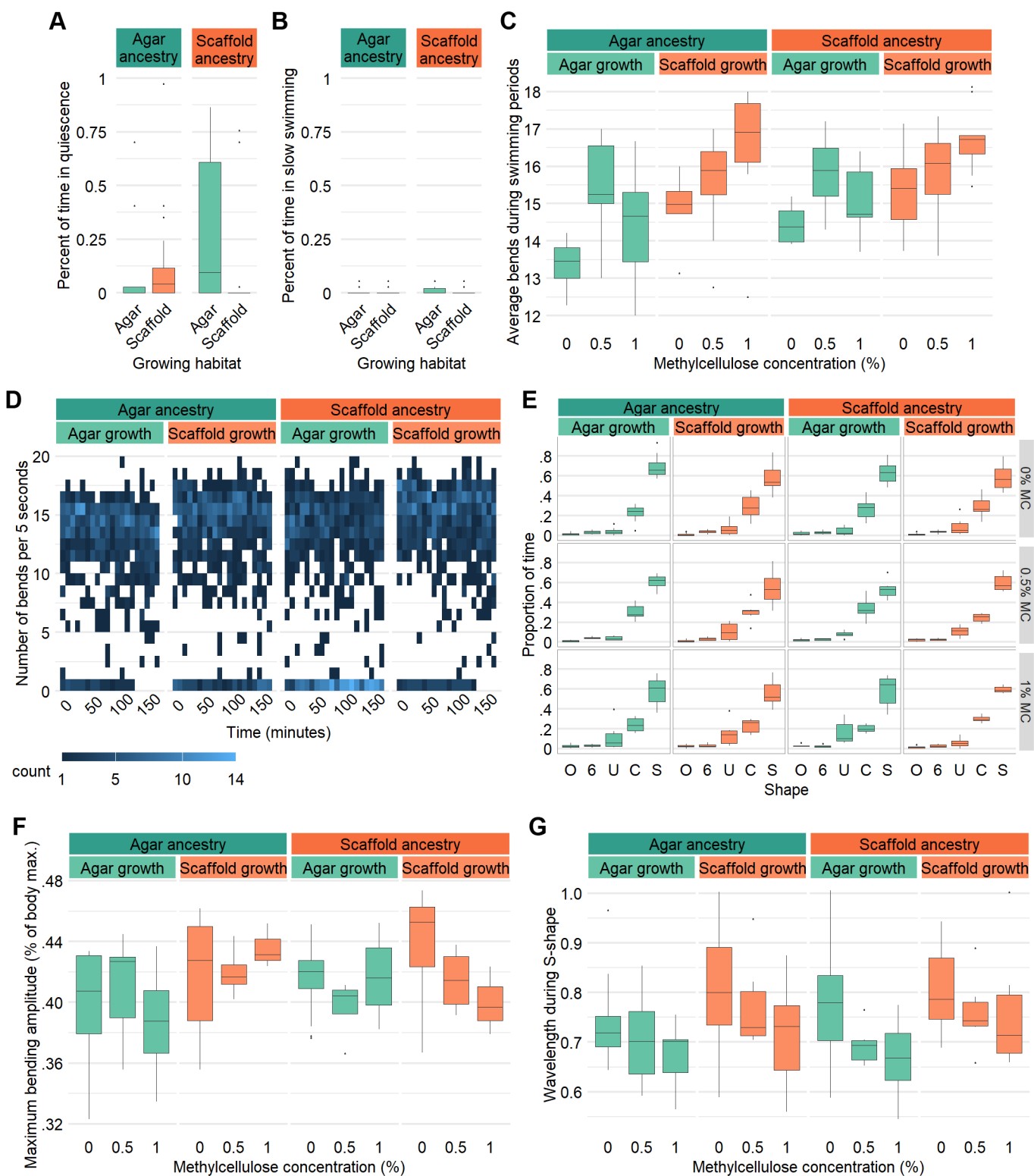

**Fig. 5. Swimming activity levels and kinematics are influenced by rearing environment and ancestry.** (A) Percentage of the 3 h experimental period spent in quiescence (≤2 bends/5 s). (B) Percentage of the 3 h experimental period spent in slow swimming (3–4 bends/5 s). (C) Average number of bends in 5 s during swimming periods (≥5 bends/5 s) for different viscosities. (D) Heatmap of number of swimming bends per 5 s through time in 0% MC. Sample size (A–D): agar:agar: 0% MC: $n$=17, 0.5% MC: $n$=11, 1% MC: $n$=9; agar:scaffold: 0% MC: $n$=17, 0.5% MC: $n$=9, 1% MC: $n$=10; scaffold:agar: 0% MC: $n$=17, 0.5% MC: $n$=11, 1% MC: $n$=10; scaffold:scaffold: 0% MC: $n$=17, 0.5% MC: $n$=7, 1% MC: $n$=9. (E) Percentage of recorded frames where the worm body was in O, 6, U, C, or S-shape by viscosity. (F) Maximum bending amplitude in proportion to body length by viscosity (0 being perfectly flat and 1 being perfectly folded in half). (G) Number of wavelengths per body length for frames in S-shape by viscosity. Sample size (E–G): agar:agar: 0% MC: $n$=15, 0.5% MC: $n$=6, 1% MC: $n$=6; agar:scaffold: 0% MC: $n$=14, 0.5% MC: $n$=6, 1% MC: $n$=6; scaffold:agar: 0% MC: $n$=15, 0.5% MC: $n$=6, 1% MC: $n$=6; scaffold:scaffold: 0% MC: $n$=14, 0.5% MC: $n$=6, 1% MC: $n$=6.

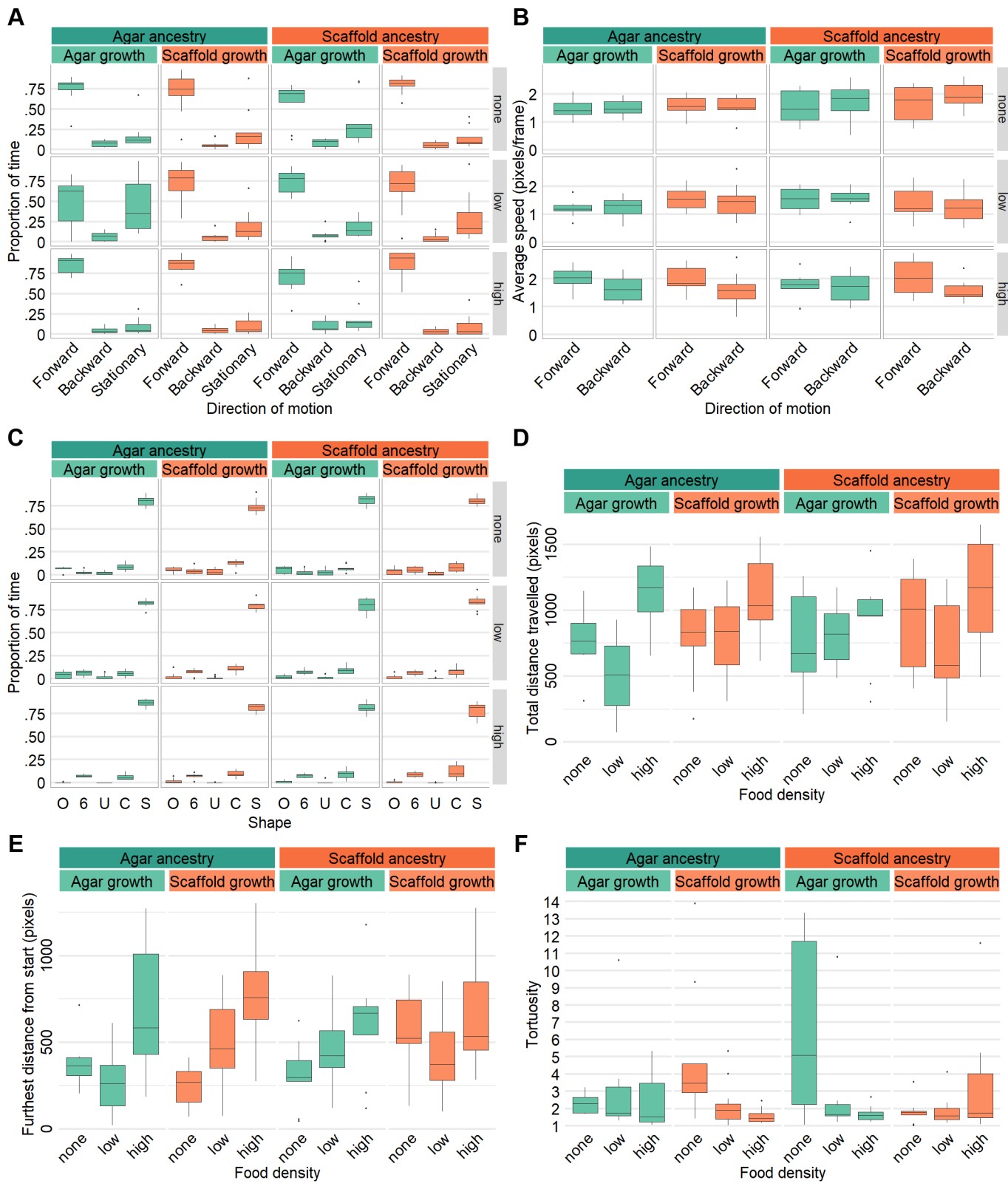

**Fig. 6. Crawling behavior is mildly affected by environmental context, ancestry, and food availability.** For 1 min recordings. (A) Proportion of time spent per recording in different directions of motion (forward, backward or stationary) by food density. (B) Average speed in pixels per frame for forward and backward movement by food density. (C) Proportion of time per recording where the worm body was in O, 6, U, C, or S-shape by food density. (D) Total distance traveled in pixels by food density. (E) Furthest Euclidean distance reached from start point in pixels by food density. (F) Path tortuosity (total distance travelled by distance between start and end points) by food density. Sample size (A–F): agar:agar: no food: $n$=8, low food density: $n$=11, high food density: $n$=10; agar:scaffold: no food: $n$=9, low food density: $n$=11, high food density: $n$=10; scaffold:agar: no food: $n$=9, low food density: $n$=11, high food density: $n$=9; scaffold:scaffold: no food: $n$=9, low food density: $n$=11, high food density: $n$=10.

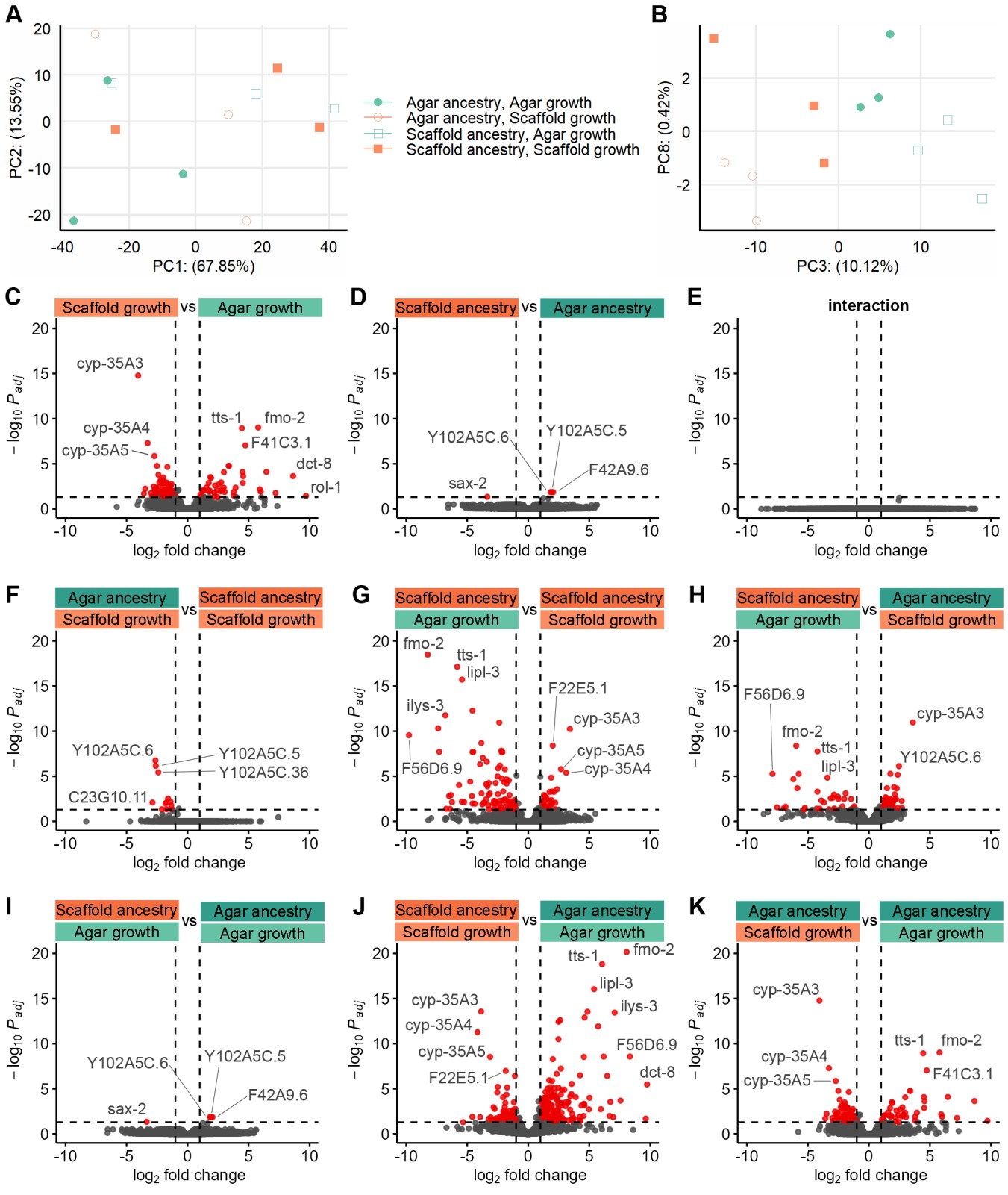

**Fig. 7. Transcriptional profiles reveal mild, rapidly reversible changes.** Principal components (A) 1 and 2, and (B) 3 and 8, where each point represents a biological replicate of RNA sequencing on young adults (three per condition). Volcano plots of factorwise differential expression analysis for (C) growth habitat, (D) ancestry habitat and (E) interaction term. Volcano plots of pairwise differential expression analysis for (F) agar:scaffold versus scaffold:scaffold, (G) scaffold:agar versus scaffold:scaffold, (H) scaffold:agar versus agar:scaffold, (I) scaffold:agar versus agar:agar, (J) scaffold:scaffold versus agar:agar and (K) agar:scaffold versus agar:agar groups. For C–K, each dot represents a gene with red-colored being significantly differentially expressed between the two conditions by a log₂fold change >|1| and a P-adjusted <0.05. Some genes of interest are labelled.

expression analysis. This analysis confirmed that the rearing habitat was the most significant factor influencing gene expression (Fig. 7C,D). We found no significant interaction effect between the rearing and ancestral habitats (Fig. 7E).

Pairwise comparisons further supported these findings with the most differentially expressed genes (DEGs) found between scaffold:scaffold and agar:agar groups, and the least between worms of matching growing habitats (Fig. 7F–K). Overall, most DEGs genes were uncharacterized or poorly studied to date.

To understand the functional implications of these transcriptional changes, we performed an over-representation analysis (ORA) using Gene Ontology (GO) slim terms, providing a high-level overview of the affected biological functions. Only seven GO slims were significantly identified across all groups (Fig. 8). All were upregulated in scaffold-grown animals except for 'organelle' and 'cell differentiation', which tended to be downregulated (Fig. 8). A complete list of ORA on granular GO terms is also available (Figs S5–S7).

## DISCUSSION

This exploratory study sought to investigate the influence of environmental context on various phenotypic and behavioral traits

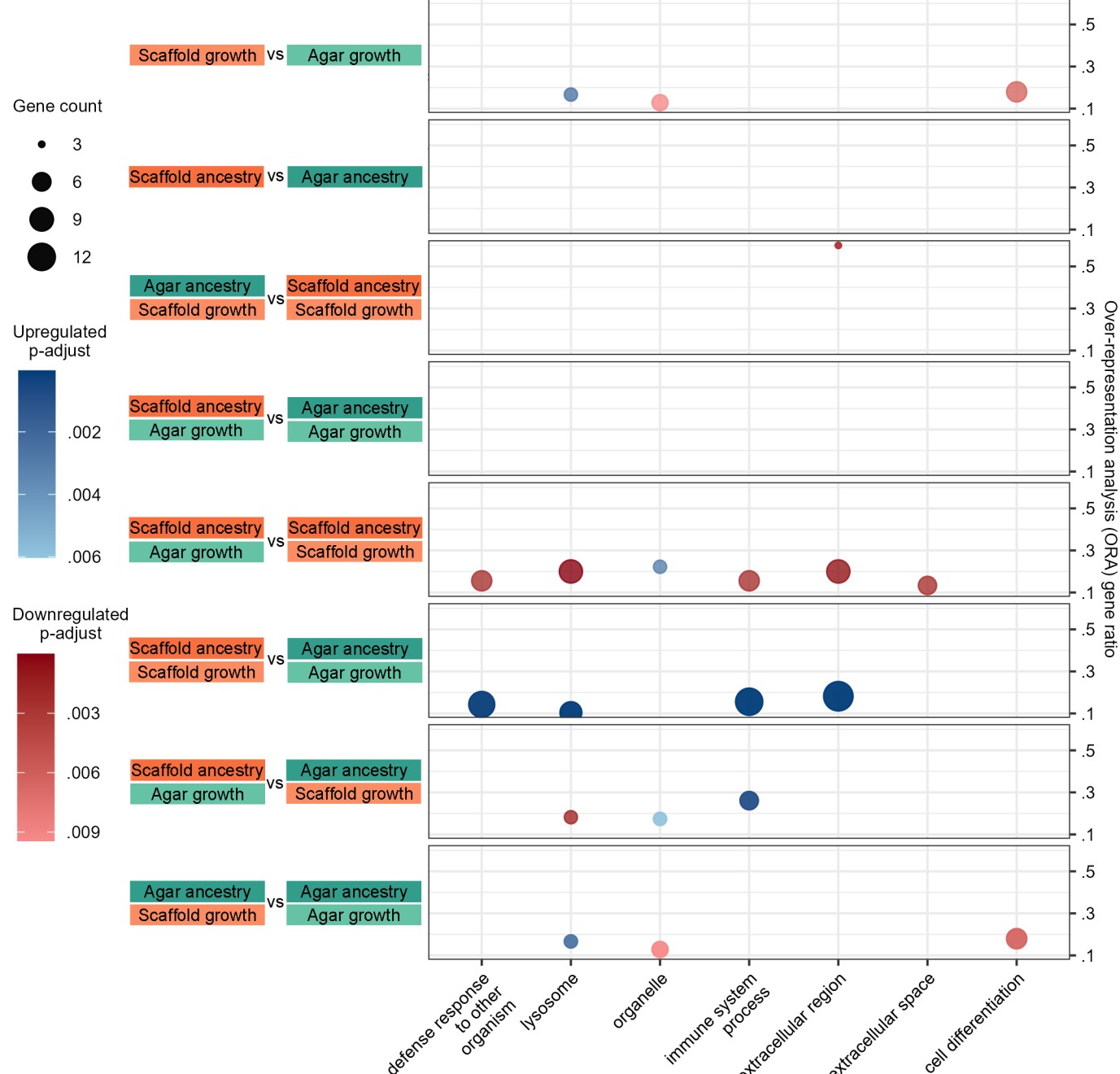

**Fig. 8. Functional analysis highlights upregulation and downregulation in scaffold-grown worms.** Dot plot of ORA of GO slim terms that were found to be significantly up- or downregulated in the identified DEGs (Fig. 7C–K) per condition. The following conditions did not have enough DEGs for ORA: ancestry factor (up- and downregulated genes), agar:scaffold versus scaffold:scaffold (upregulated genes), scaffold:scaffold versus agar:agar (downregulated genes), scaffold:agar versus agar:agar (up- and downregulated genes).

of *C. elegans*. By comparing worms cultured in 3D decellularized fruit tissue scaffolds with those raised under standard 2D agar plate conditions, we aimed to uncover how environmental context affects organismal biology and to set the groundwork for future research in this area. Our preliminary findings suggest that some biological traits of *C. elegans* are affected by the rearing habitat, supporting the notion that laboratory conditions can influence experimental outcomes and potentially our understanding of the phenomena studied.

### Global patterns of environmental responsiveness

Our compound phenotypic divergence analysis revealed that, considering our ten measured traits, the immediate rearing environment exerts a significant effect on global *C. elegans* biology, while ancestral environment shows a weaker, non-significant influence. This pattern aligns with the general principle that phenotypic plasticity allows organisms to respond rapidly to current conditions, with intergenerational effects typically being more subtle or context-dependent (Bonduriansky and Day, 2009; Uller et al., 2015; West-Eberhard, 2003). Notably, not all traits showed marked reactions to the growth environment, suggesting a hierarchy of trait responsiveness that we discuss below.

### Life history modulations

We observed that scaffold-grown worms, regardless of ancestry, had generally reduced body sizes, held a greater number of eggs in their gonads relative to their body size in adulthood, and laid more elongated eggs. Only scaffold-grown worms with scaffold ancestry had reduced total brood size. Fat content through life and adult pumping rate were not affected. In previous work, *C. elegans* has been observed to alter its growth rate and brood size with temperature, food availability, oxygen content, population density and presence of pathogens (Byerly et al., 1976; Cypser et al., 2013; Fenk and de Bono, 2017; Harvey and Orbidans, 2011; Harvey and Viney, 2007; Lenaerts et al., 2008; Szewczyk et al., 2006; Wong et al., 2020), but to much greater extremes than what has been observed here. It is crucial to emphasize that our observed life history modulations do not appear to be driven by caloric restriction (CR) as they contrast sharply with that of worms undergoing CR, and our RNA sequencing data did not reveal characteristic transcriptional profiles of CR (Fig. S8) (Bar et al., 2016; Lenaerts et al., 2008; Szewczyk et al., 2006; Zhang and Mair, 2017). Overall, our results align with the concept of developmental plasticity, where organisms adjust their life history traits in response to environmental cues (West-Eberhard, 2003). For example, it has been suggested that reduced fecundity in *C. elegans* is beneficial to the colony's overall fitness (Galimov and Gems, 2020). Our preliminary observations suggest that *C. elegans* can exhibit hierarchical plasticity in its life history in response to environmental changes without the presence of explicit stressors; some traits, such as body size, show greater flexibility and are more readily adjusted, while others, like fat content and reproductive output, demonstrate higher stability, suggesting they may be more critical to the organism's fitness and are thus more robustly maintained across different environments.

### Behavioral changes in response to environmental context

While most measured behavioral changes were unaffected, we recorded mild increases in average swimming speed, exploration distance, oxidative stress resistance, and burrowing ability in scaffold-grown animals. Our observed behavioral changes could reflect enhancement in neuromuscular abilities or altered sensory

processing. These results are similar to research in other model organisms showing that experience with a more complex environments can enhance motor skills, resistance to stressors and disease, and cognitive abilities (Chen et al., 2018; Hillenmeyer et al., 2008; Laviola et al., 2008; Mallory et al., 2016; Nithiananthharajah and Hannan, 2006; van Praag et al., 2000; Volgin et al., 2018). In *C. elegans*, swim exercise has been shown to enhance many aspects of health in aging animals (Laranjeiro et al., 2019). While more research is needed to determine how experience with the scaffold habitat affects worm health, our preliminary results suggest that the rearing environment is sufficient to affect some aspects of fitness.

### Transcriptomic profiles of environmental adaptation

Similar to the relatively subtle phenotypic changes observed, RNA sequencing revealed that the rearing environment induced only mild changes in the transcriptional profiles of young adult worms, though the relationship between these expression changes and the observed phenotypes remains to be determined. This aligns with previous work demonstrating that *C. elegans* transcriptomes are highly sensitive to laboratory conditions, with variations in temperature and diet causing significant gene expression changes (Gómez-Orte et al., 2018). The transcriptional adaptations to the environment were highly plastic, arising and reversing within a single generation, consistent with the rapid transcriptional responsiveness characteristic of this organism (Hall et al., 2010; Pradhan et al., 2019). Functionally, the DEGs in scaffold-grown animals showed an upregulation in pathways related to defense responses and immune system processes, and a downregulation in genes associated with cell differentiation and organelles. These might result from direct mechanosensory or chemical cues rather than direct pathogen recognition, reminiscent of 'surveillance immunity', which may also explain the modest enhancement in oxidative stress resistance (Ermolaeva and Schumacher, 2014; Pukkila-Worley, 2016). This analysis provides a starting point for understanding the molecular mechanisms underlying environmental adaptation and suggests that while *C. elegans* rapidly adjusts its transcriptome in response to immediate environmental context, many of the affected genes remain poorly characterized, highlighting opportunities for functional discoveries (Higgins et al., 2022).

### Intergenerational inheritance of environmental context

Some measurements differed more drastically in worms that were raised in conditions mismatched from their ancestors, suggesting a possible intergenerationally inherited influence. Particularly, both agar:scaffold and scaffold:agar worms were more quiescent when swimming and exhibited more tortuous crawling paths in the absence of food. This indicates that the environmental mismatch may trigger specific adaptive behaviors under resource scarcity that promote local search. Also, agar:scaffold worms showed reduced success in the burrowing assay, and did not have the reduced brood size observed in scaffold:scaffold worms. Yet, this group did have a generally reduced body size and a similar proportion of eggs in their gonads, suggesting that they could be laying eggs faster. These observations are consistent with studies showing that *C. elegans* can transmit information across generations (Klosin et al., 2017; Rechavi and Lev, 2017). Interestingly, two of the only four DEGs by ancestry (Y102A5C.5 and Y102A5C.6) are pseudogenes that have been found to be involved in transgenerational inheritance and learning (Posner et al., 2019; van der Linden et al., 2010). This suggests that the observed behavioral inheritance might be mediated specifically through small RNAs and post-translational modifications,

which are not detected by traditional mRNA sequencing. Nonetheless, the rapid reversal of the vast majority of gene expression changes and phenotypes indicates that most of the observed effects are primarily due to direct environmental influences rather than intergenerationally inherited modifications. This capacity for rapid adaptation and equally rapid reversal is consistent with the natural boom-and-bust, transient lifestyle of *C. elegans* (Frézal and Félix, 2015).

Collectively, our findings suggest that *C. elegans* could be exhibiting hierarchical plasticity in response to environmental complexity, with some traits (body size, certain behaviors) showing greater flexibility while others (fat content, feeding rate) remain robustly maintained. This hierarchy could reflect the relative importance of different traits to fitness where core metabolic functions are buffered against environmental variation, while other developmental and behavioral parameters retain plasticity to optimize performance in local conditions (Stearns, 1989). These findings could mean that phenotypes measured in standard 2D conditions may represent only a subset of the organism's biological repertoire, potentially affecting the generalizability of findings (Barrière and Félix, 2005). The dominance of immediate environmental effects over ancestral ones observed in our divergence analysis is consistent with the natural ecology of *C. elegans*, where rapid phenotypic adjustments to current conditions are favorable for survival (Frézal and Félix, 2015). Furthermore, our observation that environmental mismatch between generations triggers specific behavioral responses (increased quiescence and tortuosity) highlights a rarely considered variable in experimental designs. These phenotypes were not intrinsic to either habitat but emerged specifically from intergenerational change, a distinction that single-generation studies miss entirely. This reinforces the value of multigenerational acclimation, as responses to novelty or change can be confounded with stable adaptations (Cheung et al., 2005; Pradhan et al., 2019; Swierczek et al., 2011; Weber et al., 2010).

As an exploratory study, the sample sizes for certain assays may not be sufficient to draw definitive conclusions; larger cohorts would increase statistical power and confidence in the results. While the scaffold habitat provides structural enrichment, other environmental factors, such as differences in local food availability, mechanosensory feedback, oxygen diffusion, or waste accumulation, could be contributing to the observed effects. Controlling for these factors or understanding their dynamics in the scaffold would strengthen the causal links between environmental complexity and phenotypic outcomes. For example, variations in the occupancy task (Guisnet et al., 2021a) could help elucidate which characteristics make the scaffold habitat attractive. In addition, while the similar timing in egg production suggests similar larval development progression between conditions, the proportion of larval stages between conditions need to be explicitly quantified. Also, beyond the identified DEGs, the specific molecular pathways mediating the observed phenotypic changes require further investigation through targeted genetic manipulations, epigenetic modifications, and the study of inherited small RNAs and proteomic profiles. Likewise, a large inventory of additional physiological parameters still remains to be evaluated.

In conclusion, this exploratory study provides evidence for the impact of environmental context on *C. elegans* biology, challenging the notion that findings from simplified laboratory conditions can fully capture the organism's biological potential. By bridging the gap between laboratory and more naturalistic conditions, this work sets the stage for a more nuanced understanding of gene-environment interactions in *C. elegans* and potentially other model organisms. Future research building on these findings could lead to significant advances in our understanding of phenotypic plasticity, adaptation, and the interplay between genes and environment in shaping biological outcomes.

## MATERIALS AND METHODS

### Strains and culture

Wild-type N2 *C. elegans* provided by the CGC were grown under standard conditions at 20°C and were fed OP50 *E. coli* bacteria (Brenner, 1974).

Scaffold and traditional agar plate habitats were created as described in Guisnet et al. (2021b) without modification. In short, apple slices are decellularized using SDS and subjected to several washes. Scaffolds and agar plates are then incubated with nematode growth medium *E. coli* culture to ensure uniform growth throughout.

For all data reported, animals were first thawed from fresh wild-type N2 stock and maintained for ten generations on traditional agar plates. Then, maintenance on scaffold plates started in parallel for at least ten generations before experimental testing. Animals were maintained by moving three young adults to fresh plates every 3 days. For testing intergenerational inheritance, three young adults were moved to a fresh plate opposite of their own growing habitat. These offspring were thus grown from birth in the environment mismatched from their ancestors (10+ prior generations).

### HR-MAS nuclear magnetic resonance spectroscopy

The scaffolds were prepared as described in Guisnet et al. (2021b) up until storage at 4°C. 30 samples were taken out of five different stored scaffold slices using a biopsy punch. The samples were left to vacuum dry for 5 h at room temperature and then rehydrated in deuterated water ($D_2O$, D 99.9%).

HR-MAS NMR experiments were performed on a Bruker Avance III HD spectrometer operating at a resonance frequency of 599.90 MHz for 1H. The instrument was equipped with a 4 mm HR-MAS dual inverse 1H/13C probe with a magic angle gradient. All experiments were carried out at a magic angle spinning rate of 5 kHz and a temperature of 298 K.

Bruker TOPSPIN software (version 3.6, patch level 5) was used to acquire and process the NMR data. The 1D 1 H HR-MAS NMR spectra were recorded using a 1D NOESY two-step presaturation sequence for water suppression ("noesypr1d" from the Bruker pulse-program library). Each 1D 1 H NMR spectrum consisted of co-adding 256 transients with a spectral width of 12 kHz, a data size of 12 K points, an acquisition time of 0.5 s, and a relaxation delay of 2 s. The coadded free induction decays (FIDs) were exponentially weighted with a line broadening factor of 0.3 Hz, Fourier-transformed, phase and (polynomial) baseline corrected to obtain the 1 H NMR spectra.

### Pumping rate

Pumps of young adults were counted directly on maintenance plates at room temperature on a compound microscope. 10 min before counting, three 2 µl drops of IAA diluted 1:100 in double-distilled water were added to the lid of the plates to encourage movement to the surface. The experimenter counting the pumps was blind to the worms' ancestry.

### Developmental timing, gonad egg count and egg size

Worms were timed by placing egg-laying adults on plates for 2 h. These offspring were moved to fresh plates through the experimental days as needed to keep them fed. Every 24 h after being laid, worms were placed on 2% agarose pads in a 5 µl drop of 1 M sodium azide to immobilize them. After 1 min, the sodium azide was carefully absorbed with a Kimwipe. No coverslip was added to cover the worms as this caused them to be squished and deformed (particularly in width). Worms were imaged within 10 min of being immobilized. Grayscale images were taken with an Olympus XM10 CCD camera mounted on an Olympus MVX10 microscope using an MV PLAPO 2XC objective. TIF images were acquired with a 1376×1038 resolution and 10 ms exposure with the cellSens software (Evident).

Starting on day 3, worms were too thick to be completely in focus. Thus, two images were taken for each animal: one with the head and tail in focus, and a second with the mid-body in focus. On day 5, three adults from three different plates for each condition were moved to an empty agar plate to lay

eggs for 2 h. Eggs were imaged directly on the agar plate under the same acquisition settings as described above.

Worm and egg images were segmented as described in Guisnet and Hendricks (2025). For each worm, we extracted the following morphological features: area (sum of pixels within the worm), perimeter (Euclidean number of pixels), length (pixel count of the worm medial axis), width (estimated at each point along the medial axis). Egg elongation was determined by fitting an ellipse to the egg contour and calculating the ratio of the major axis to the minor axis.

The number of eggs present in the worm gonads were counted using the images that were focused on the midbody of the worm by a *C. elegans* expert blind to the condition using the LabelBox web tool with academic access (*Labelbox*, 2025).

### Oil Red O fat staining

Oil Red O staining was done following the protocol described in He (2012) except that instead of using a thermomixer shaker, tubes were taped to a box filled with room-temperature heat beads and shaken at 150 RPM. The high velocity of the shaking platform created rigorous movement on the top beads which collided with the bottom of the tubes, recreating the effect of the shaker. Over a 5-day period prior to staining, one egg-laying adult was moved daily to maintenance plates to allow for longer growth without food depletion. For each condition, worms were pooled from all the differently aged plates to obtain a mix of stages. Colored images were taken with a QImaging MicroPublisher RTV 3.3 camera with a Leica 10445930 1.0× mount lens. TIF images were acquired with a 2048×1536 resolution, 75 ms exposure and 24-bit depth. Worms were segmented as described in Guisnet and Hendricks (2025). Worm area, perimeter, length, and width were acquired as described above for developmental timing measurements.

For background correction, a composite background mask was created from all 'background' classified areas. Fifty random background pixels were selected, each at least 100 pixels away from image edges and non-background areas. Mean red, green and blue (RGB) values of these pixels were calculated. RGB channels of the worm pixels were then normalized against these background values.

For staining measurement, more intensely stained (fat) locations are more intensely red, but also darker as the intensity of the dye decreases the passage of light. Hence, a staining metric was calculated for each worm pixel by multiplying the red dominance and the inverse luminance. The luminance is the normalized red channel value, representing the 'brightness' of the pixel – more stain means less light passing through and a lower normalized channel value. Red dominance is the intensity of the red channel compared to the green and blue channels (red_dominance = normalized_red − (normalized_green + normalized_blue)/ 2). More uniform channels show as a whiter color. The higher the discrepancy between the channels, the more the color stands out. Using this composite staining metric ensures we are capturing both the amount of staining in the worm body, but also the intensity of that staining. Overall, worm staining was calculated as the average staining metric per pixel.

### Brood size

The same protocol as described in Guisnet et al. (2021b) was followed to create both habitats, with the following modifications: 4 cm diameter plates were used, scaffold were cut to 1.5 cm diameter, 10 µl of NGML *E. coli* culture was added directly on both types of plates, plates were parafilmed and kept upside down in a closed box 24 h prior to use.

Worms were synchronized and placed on individual plates at the late L4 stage. Every 24 h, for 4 days, worms were moved to a fresh plate to distinguish egg-laying days. Worms that died or were found to be in plates with contamination were completely removed from the dataset. Offspring were counted 3 days after being laid. Scaffold-born offspring were lured out of the scaffold with food and manually extracted from the scaffold by shaking in liquid.

### Burrowing assay

We performed the burrowing assay as described in Laranjeiro et al. (2019) with minor modifications. We used 96-well plates and added only one worm per well. 5 µl of LB *E. coli* OP50 culture was used as the attractant at the top

of the pluronic gel. Young adult worms were selected from maintenance plates and left to roam off food for 2 min before being added to the well.

### Oxidative stress assay

Young adult worms were selected from maintenance plates and moved to empty agar plates to crawl off food for 5 to 10 min. 180 µl of 3 mM $H_2O_2$ was added to each well of a 96-well plate and one worm per well was transferred with a platinum worm pick. Worms were checked every 2 h for 8 h. Before marking an inactive worm as dead, it was gently poked with a platinum worm pick. Animals were also checked at time 0 (right after being added to the well). Animals dead at time zero were removed from the dataset.

### Long-term swimming bends count

This protocol was inspired by Ghosh and Emmons (2008). Young adult worms were picked from maintenance plates and left to roam off food for 10 min on an empty agar plate. They were then transferred to individual wells of a 96-well plate filled with 180 µl of room temperature filtered NGML. Counting started within 10 min of the first worm being transferred. The number of bends made in 5 s was counted by eye with an auditory timer every 5 min over 3 h. Worms that were found dead at the end of the experimental period were removed from the dataset.

For high viscosity swimming, 0.5% and 1% methyl cellulose (MC) in filtered NGML was prepared by serial dilution. Their viscosity was verified with a RheoSense m-VROC II viscometer to be 5.6 cP (similar to milk) and 20 cP (similar to vegetable oil), respectively. The viscosity of filtered NGML was verified to be 1 cP (water).

### Short-term droplet swimming recording and analysis

50 µl of filtered NGML or 3.5 µl of MC solution was pipetted onto the center of a 10 cm diameter plexiglass dish. 3.5 µl was used for the MC solutions as the higher viscosity caused the droplets to stay relatively spherical and thick, leading to worms often swimming out of focus or turning on the z-axis. Thus, the MC droplets were flattened, but kept in a round shape, by strongly smashing the plexiglass dish onto the experimenter's palm. Young adult worms were selected from maintenance plates and left to roam off food on an empty agar plate for 2 min before being moved to the droplet with a platinum worm pick.

AVI videos were recorded for 1 min in NGML and 30 s in MC solutions at 10 fps with a FLIR Blackfly (BFS-U3-51S5M-C) 5.0 MP camera and Spinnaker SDK software (Teledyne) at 2448×2048 resolution. The recording plates were held upside down with a custom rig and recorded from below. An LED light ring of the same size as the plate was positioned centered with the plate, but further up (∼ 8 cm), to minimize light refraction artifacts. The recordings were analyzed as described in Guisnet and Hendricks (2025).

### Crawling recording and analysis

Young adult worms were selected from maintenance plates and left to roam off food on an empty agar plate for 2 min. 10 cm diameter NGM agar plates were used with either nothing (no food), 300 µl of NGML OP50 *E. coli* culture added 1 h before recording (low food density) or 300 µl of NGML OP50 *E. coli* culture incubated at 37°C for 24 h and acclimated at room temperature prior to recording (high food density).

AVI videos were recorded for 1 min as described for swimming and analyzed as described in Guisnet and Hendricks (2025).

### RNA sequencing and data processing

Young adult *C. elegans* were collected by hand from maintenance plates directly into 25 µl of TRIzol reagent. Samples were stored at −80°C until sufficient quantities were collected.

Frozen samples were vortexed for 15 min and 150–250 worms were pooled for each sequencing sample. These pooled samples underwent three freeze-thaw cycles: freezing at −80°C for at least 20 min, followed by 15 min of vortexing before final storage at −80°C until sequencing.

Samples were further processed and sequenced by the GenomeQuébec genomics facility. Total RNA was extracted and sequenced using Illumina NovaSeq 6000 platform, generating paired-end reads of 100 bp. Three

biological replicates per condition were sequenced, with each replicate receiving 2×25 million reads.

Sequencing reads were cleaned and aligned to the *C. elegans* reference genome using a custom Linux shell script. Quality control of raw sequencing data was performed using FastQC (v0.11.9) and MultiQC (v1.12) (Andrews, 2010; Ewels et al., 2016). Reads were trimmed using Trimmomatic (v0.39) to remove low-quality bases and potential adapter contamination (Bolger et al., 2014). Specifically, the first 10 bases were removed from the 5′ end of each read to mitigate potential priming biases, and a sliding window quality trimming was performed (window size 4, required quality 15). Trimmed reads shorter than 36 bp were discarded. The trimmed high-quality reads were then aligned to the *C. elegans* reference genome (WBcel235) using HISAT2 (v2.2.1) with default parameters (Kim et al., 2019). Post-alignment, a minimal filtering step was performed to remove a small number of malformed alignment entries (31 out of 161,930,474 total entries). The resulting alignments were converted to BAM format, sorted, and indexed using SAMtools (v1.13) (Danecek et al., 2021). Gene-level quantification was performed using featureCounts (v2.0.3) against the Ensembl gene annotation (release 104) (Liao et al., 2014). Further analysis was performed in R (v4.3.2) (R Core Team, 2021).

Gene names, GO terms, GO slim terms and all their corresponding descriptions were fetched from WormBase BioMart with the biomaRt package (v2.58.2) using the "celegans_gene_ensembl" dataset (Durinck et al., 2009; Harris et al., 2020; *WormAtlas Navbar*, 2002-2024). Genes were matched to GO terms with the org.Ce.eg.db package (v3.18.0) (Carlson, 2023).

Differential expression analysis was performed with the count data using the DESeq2 package (v1.42.1) (Love et al., 2014). A DESeqDataSet object was created using the count data and condition labels (pairwise and factor-wise), and differential expression analysis was conducted using the DESeq function with default parameters. PCA data were generated with the DESeq2 plotPCA function. Volcano plots were generated with the EnhancedVolcano package (v1.20.0) (Blighe et al., 2023). Over-representation analysis was conducted on GO and GO slim terms with enrichGO and enricher function of the ClusterProfilter package (v4.10.1) with default parameters (Wu et al., 2021).

## Statistical analysis

Image analysis was conducted in Python (v3.12.3), and all data manipulation and visualization were conducted in R (v4.3.2) with the tidyverse (v2.0.0) and ggplot2 (v3.5.2) (R Core Team, 2021; *The Python Language Reference*, 2024; Wickham, 2016; Wickham et al., 2019). All boxplots represent the interquartile range (IQR), with the median indicated by the horizontal line. Whiskers extend to the furthest data point within 1.5 times the IQR. Outliers beyond this range are shown as individual points. Scatterplots have all data points shown. The line and point plots have error bars representing standard error calculated at each time point. All experiments were replicated at least three times in the laboratory, with all conditions in parallel.

Instead, we quantified the overall phenotypic impact of the growth environment and ancestry by computing a compound $D^2$ that integrates effect sizes across phenotypic modules. We pre-defined ten biological modules: feeding rate, body morphology including gonad egg count, fat content, reproductive output, egg morphology, burrowing ability, oxidative stress resistance, swimming behavior, crawling behavior, and transcriptomic profiles. For each module, we fit either ANOVA (single-trait modules), MANOVA (multi-trait modules), or linear mixed-effects models (repeated measures) with rearing environment (agar versus scaffold), ancestry (agar versus scaffold), and their interaction as predictors (2×2 factorial design). From each model, we extracted $t^2$-equivalent effect sizes. The global $D^2$ statistic for each factor was computed as the sum of $t^2$ values across all modules and thus represents the effect size for each factor. Statistical significance was assessed via permutation tests (5000 iterations). For each permutation, factor labels were shuffled within the data for each module, models were refit, and $D^2$ was recomputed. Permutation $P$-values were calculated as the proportion of permuted $D^2$ values higher than the observed $D^2$ values.

By not computing $P$-values at the trait level, we aimed to minimize the risk of Type I errors that can arise from multiple comparisons in exploratory

analyses. This strategy reduces the likelihood of identifying spurious effects that may occur by chance when testing numerous variables. Furthermore, avoiding $P$-value calculations helps prevent the misinterpretation of statistical significance in a context where findings are preliminary and require validation in future confirmatory studies (Makin and Orban de Xivry, 2019).

## AI disclosure statement

AI tools were used to accelerate code production and improve writing quality.

## Acknowledgements

We thank Stephanie C. Weber from McGill University for sharing their colored camera for Oil Red O imaging. Christopher Moraes and Claire Edrington for the viscosity measurements. Giovanni Vizcardo for helpful insights on worms in viscous liquids. GénomeQuébec for the library preparation and RNA sequencing. André Gravel at the Research Institute of the McGill University Health Centre and Alexandre Arnold at the Department of Chemistry at Université du Québec à Montréal for performing the HR-MAS analysis. WormBase and WormAtlas teams for providing invaluable resources. All members of the lab for helpful discussions and comments. Strains were provided by the *Caenorhabditis* Genetics Center (CGC), which is funded by NIH Office of Research Infrastructure Programs (P40 OD010440).

## Competing interests

The authors declare no competing or financial interests.

## Author contributions

Conceptualization: A.G., M.H.; Data curation: A.G.; Formal analysis: A.G., M.R.; Funding acquisition: A.G., M.H.; Investigation: A.G., N.H., B.R.Q.; Methodology: A.G., M.H.; Project administration: M.H.; Software: A.G., M.R.; Supervision: M.H.; Writing – original draft: A.G.; Writing – review & editing: A.G., M.H.

## Funding

This work was supported by funding from the National Science and Engineering Research Council (NSERC) (RGPIN/05117-2014); the Canadian Foundation for Innovation (CFI) (32581); the Canada Research Chairs Program (950-231541); and the Fonds de Recherche du Québec – Nature et Technologies (300853 to A.G.). The funders had no role in study design, data collection and analysis, decision to publish, or preparation of the manuscript. Open Access funding provided by McGill University. Deposited in PMC for immediate release.

## Data and resource availability

The data discussed in this publication have been deposited in NCBI's Gene Expression Omnibus (Edgar et al., 2002) and are accessible through GEO Series accession number GSE317801 (https://www.ncbi.nlm.nih.gov/geo/query/acc.cgi?acc=GSE317801). All other datasets are available on FigShare (10.6084/m9.figshare.30071044). All relevant data and details of resources can be found within the article and its supplementary information.

## Peer review history

The peer review history is available online at https://journals.biologists.com/bio/lookup/doi/10.1242/bio.062282.reviewer-comments.pdf

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
