## [Peer Review File · Biology Open]

The impact of rearing environment on *C. elegans*: Phenotypic, transcriptomic and intergenerational responses to 3D enriched habitats

Aurélie Guisnet, Nour Halaby, Maxime Rivest, Beatriz Romero Quineche and Michael Hendricks

DOI: 10.1242/bio.062282

Editor: Sandhya Koushika

Review timeline

Original submission:	23 September 2025
Editorial decision:	13 October 2025
First revision received:	21 January 2026
Accepted:	26 January 2026

Original submission

First decision letter

MS ID#: bio.062282

MS Title: The impact of rearing environment on *C. elegans*: Phenotypic, transcriptomic and intergenerational responses to 3D enriched habitats

Authors: Aurélie Guisnet, Nour Halaby; Maxime Rivest; Beatriz Romero Quineche; Michael Hendricks

I have now reached a decision on the above manuscript.

The reviewer reports are shown at the bottom of this email or can be accessed, together with a copy of this decision letter, by going to:

As you will see, the reviewers raised a number of substantial criticisms that prevent me from accepting the paper at this stage.

It would help the manuscript if you could frame the scientific question that is being addressed despite it being an exploratory study in the appropriate places in the manuscript. It would also help immensely if some statistical analyses could be presented.

The reviewers suggest, however, that a revised version might prove acceptable, if you can address their concerns. If you think that you can deal satisfactorily with the criticisms on revision, I would be pleased to see a revised manuscript. We would then return it to some the reviewers.

At this stage, we also ask you to ensure your manuscript complies with our formatting guidelines. Provided you are able to fully address the referees' comments, we are positive about publication of your paper (we accept over 95% of revision submissions) and therefore hope you won't mind any extra work involved in reformatting your manuscript at this point.

Please upload both a 'clean' version of your Word file, along with a highlighted version clearly showing where you have made changes in the revised manuscript. Please avoid using 'Track changes' in Word files as these are lost in PDF conversion.

I should be grateful if you would also provide a point-by-point response detailing how you have dealt with the points raised by the reviewers in the 'Response to Reviewers' box. Please attend to all of the reviewers' comments. If you do not agree with any of their criticisms or suggestions please explain clearly why this is so.

Reviewer 1

Comments for the author

In this paper, Guiset et al. examine how rearing environments - standard two-dimensional agar plates versus three-dimensional fruit tissue scaffolds - influence *C. elegans* biology. They report that while fat content and feeding rate remained unchanged, scaffold-grown worms exhibited smaller body size and reduced brood size. Additionally, these worms showed modest improvements in stress resistance, burrowing, swimming, and exploratory behaviors. The altered rearing conditions also led to mild, non-heritable changes in gene expression profiles.

Overall, this is an interesting and well-characterized study of worm behavior across different environments. This study emphasizes the need to consider rearing conditions when interpreting laboratory results. However, the authors should address the following concerns to make their findings more convincing and conclusive.

Major comments:

1. In lines 47-57 (page 3), the authors test the potential toxicity of the 3D scaffold by monitoring the presence of SDS using HR-MAS technology. However, this section should also provide details about the scaffold's composition and explain the rationale for checking SDS contamination.
2. In Figure 1A and the corresponding results described in lines 21-31 (page 4), no statistical analysis has been performed to support the conclusion regarding differences in pumping rate across rearing conditions. Including appropriate statistical tests is essential to validate this claim.
3. In the section titled "*Developmental timing is altered by experience with the enriched environment*", the results presented primarily concern morphological traits - body length, body width, area, and perimeter - rather than developmental timing. Did the authors examine whether worms progressed through the developmental stages (L1, L2, L3, L4, and adult) at the same rate or over different durations? If so, those results should be reported here. The current data would be more appropriately placed under a heading such as "*Morphological alterations observed under different rearing conditions.*"
4. In Figure 1B, the authors normalized body length by body width, which represent two distinct morphological parameters. The rationale for this choice of normalization is unclear. It would be more informative to present how body length changes over time independently of body width.
5. In Figures 2 and S2, no statistical analyses are shown. Including appropriate statistical tests is necessary to support the conclusions drawn.
6. From Figures 2 and S2, Guiset et al. conclude that scaffold-reared worms are thinner and shorter, with reduced area and perimeter compared to agar-grown worms. Could these differences arise from mechanical compression within the scaffold or from reduced nutrient availability? Did the authors test these possibilities?
7. In Figure 2E, it is unclear what each data point represents. Does each point correspond to the Oil Red O staining intensity of an individual worm? If so, under the scaffold:scaffold condition, only two worms appear to have been measured within the 0-2 body length range. The sample size here is very low, and the authors should consider increasing the number of worms analyzed.

Additionally, please clarify whether the staining intensity is normalized relative to body size for each worm.

8. In lines 18-20 (page 6), the authors state that “eggs from scaffold-grown worms showed greater elongation, although variability was large across all groups.” However, Figure 3F does not display any data to illustrate this variability. The authors should either include the variability in the figure or clarify how it was assessed.

9. In lines 5-6 (page 7), the authors state that “only scaffold:scaffold animals were still alive at the 6-hour time point.” Please include the exact percentage of animals that remained alive at this time point for clarity.

Throughout the manuscript, including in this section, the authors describe the observations from their experiments but often do not provide clear inferences or conclusions drawn from these results. For example, what does the increased survival of scaffold:scaffold animals imply about the role of the scaffold environment? Adding interpretive conclusions alongside the reported data would make the findings more meaningful and strengthen the overall impact of the study.

10. In lines 4-6 (page 8), the authors state that “we calculated their bend amplitude as a proportion of their body length.” This would be clearer if the authors included a small schematic in Figure 5F to illustrate how bend amplitude was measured.

11. In lines 37-39 (page 8), the authors describe the different shapes adopted by worms (O, U, and 6-shapes). Including representative images of these shapes would be helpful.

12. For the transcriptional profiling experiments, please provide details on the number of worms used, their developmental stage, and the number of biological replicates included in the RNA sequencing.

Minor comments:

1. Please ensure that figures are presented in the order in which they are cited in the text. For example:

- Page 4: Figures S2C and S2D appear before S2B.
- Page 6: Figure 4B is cited before 4A.
- Page 7: Figure 5C is cited before 5B.

2. Figure 3E is missing from the manuscript text and should be referenced appropriately.

Reviewer 2

Comments for the author

The manuscript by Guisnet et al. focuses on exploring the influence of rearing environments on *Caenorhabditis elegans*, comparing worms raised in three-dimensional decellularized fruit tissue scaffolds with those maintained on standard agar plates. The study aims to evaluate phenotypic, behavioral, and transcriptomic differences and assess potential intergenerational effects. The topic is conceptually interesting and relevant to discussions of environmental context in model organisms. The major limitation is the deep exploratory nature of the study, which the authors have acknowledged. However, I would share my opinions as below:

1. The use of decellularized fruit tissue scaffolds introduces multiple potential confounders that are not adequately controlled or quantified in the manuscript. Without these controls, the observed effects cannot be confidently attributed to environmental enrichment per se.

2. The authors have mentioned that the p-values were not computed due to the exploratory nature of the study but even in this case, descriptive statistics accompanied by measures of uncertainty are essential to support interpretation and distinguish true effects from random variation.

3. Although RNA-seq data are presented, the current results are insufficient to connect gene expression changes with observed phenotypes in a mechanistic manner.
4. The discussion overextends the findings by implying intergenerational inheritance and environmental adaptation, though the data show only mild, reversible effects which makes the interpretations speculative and hence are not substantiated by the presented results.
5. I would also suggest adding Figure numbers on the figure itself for better quality.

Conclusively, the manuscript in its current form cannot be accepted.

Reviewer 3

Comments for the author

This paper aims to examine how the rearing environment affects a range of phenotypic and behavioral traits in *C. elegans*. By comparing worms grown in enriched scaffolds with those maintained under standard laboratory conditions, the authors explore how environmental context shapes biological processes. They further investigate whether the resulting phenotypic changes are heritable. The omnibus paper is generally well written and presented, covering a large collection of methods and results, including life history traits, behaviours, physiological parameters, transcriptome profiles, and measures of intergenerational inheritance. In the review, below, I first outline some major concerns and following this I provide some more detailed feedback with minor considerations for the author. I hope that my comments help the authors to improve their work, and I congratulate them on a very large collection of work, which undoubtedly took a lot of effort.

MAJOR COMMENTS

Firstly, the authors frame this work throughout as an 'exploratory study'. Given this, they do not provide any real research questions, hypotheses, or statistical tests. Thus, the paper is essentially a presentation of raw data for a wide range of traits that hint at trends but offers little in the way of concrete findings. I still believe this to be a useful contribution and looking at the sample sizes (somewhat hidden in the figure legends) then it seems that although more complex statistical models may not be viable, at least some simple (even if univariate) statistical comparisons may be feasible. The authors claim to be reporting 'effect sizes', but I see nothing clearly denoting this in the paper.

Secondly, and related to the above, the introduction is poorly focussed and brief. I would like to see more engagement with what the paper really aims to do (beyond simply presenting data), and what the potential applied/translational outcomes of these findings might be. There are only a handful of references in the introduction, showing quite a limited engagement with what is surely a plentiful literature on the broader questions. Given what is known, how might you expect your measured variables to vary across different rearing environments? How might this differ between behaviour, physiology, and genetic factors? I think the broad categorisation used in the discussion is useful for presenting the work clearly - i.e., headings in bold, and you might consider using these throughout aims/methods/results.

Thirdly, although well written, the paper is structured in quite an awkward way for the reader. Some of this might be more common in some sub-fields than others, however, I find having a results section which includes both detailed methods and a fair amount of interpretation to be a complex way of presenting the paper. I do understand that there are very many different tests and variables, so a classic methods and results set up might be difficult (as it would be hard to track which methods go with which results). However, I think some careful consideration of the structure is needed to ensure consistency and clarity.

Finally, much like the discussion, the discussion is a bit vague, and the authors sit on the fence with the data without really engaging with questions or applications. I respect their acceptance of a lack of statistical power - and fully support careful interpretation, but I think more can be said and

interpreted and that the authors might take a bit more of a (careful) position in what the paper is about, what the findings are, and (importantly) what the findings might mean. If a bigger study were to hold up your findings, then what would it tell us, and how could this be used?

MINOR COMMENTS

Continuous line numbers are much easier for reviewers, rather than by page.

Introduction:

In general, very clearly written, but I think a much deeper engagement with the literature is required.

P2; L23-27: this could be clearer and better framed in more specific applied research (e.g., do you mean some importance for biomedical models, behavioural models, etc.)

P3; L23-37: here I would like to see some more specific aims. First, I recommend making clear the gap in knowledge, by expanding on the literature used in the intro. For example, what is known in this species, and what about other model species? has anyone ever looked at heritability? How is this work different? Then I think you need to be clear about your contribution and its potential relevance.

P3; L23-37: the aims need to clearly set up the general variables used, e.g., refer to behaviour, life history, heritability, genetics...otherwise the methods seem to come out of nowhere.

Results:

This is more of a decision for the Editor/journal, but I found all the methods and discussion embedded into the results section confusing, as per my previous comment. Thus, I think the structure could be improved. I have not marked all the cases of what are clearly methods/discussion throughout, as it is pervasive - so if you/editor agree with any restructure then please check carefully.

P3; L47: 'the' potential toxicity?

P3; L49: what is toxic about the preparation process?

P4; L27-31: this is discussion - try and keep any interpretation for the discussion section.

P5: is habitat experience the same thing as environmental context? Clarify and if so ensure consistency.

P5; L25-27: this is discussion. In most fields, citations are avoided in results.

P5; L49-51: this is discussion.

P7; L36: 'ancestors particularly had' reword?

Discussion:

Generally good, and better framed in the literature. However, please see major comment. I like the framing around broad categories for the variables, and as per previous comments I think this could be expanded throughout.

P11; L11-29: this section needs references to published literature.

P12; L52-58: As per my previous comment, this final section needs to refer back to what your aims are, make some conclusions about what your findings mean (in more concrete terms) and make some effort to say what their relevance might be for future study or application.

Methods:

Are very well written and clear. Perhaps they could be organised around the broad headings previously mentioned?

P18:L27- here I think some more effort could be made to test some hypotheses. I understand the limitations of sample size, but I think you either need to accept that more work is needed to increase these, or have a go at interpreting what you can with the limited data.

P18; L42-44: what does 'focussing on estimating effect sizes' mean, and is it part of a statistical analysis? I don't see anything reported on this, e.g., odds ratios, Cohen's d, etc.

P18; L46: I don't see much value in having an aim to generate hypotheses in published work, and I don't see that you follow this through. Similarly, do you 'uncover novel insights without the constraints of predefined hypotheses? I am not sure I agree with the general approach. You must have had some thoughts about how the environment would impact on the variables?

Figures/legends:

Might the sample sizes be better reported in the methods section? (or repeated)

The figures are beautiful and very clear overall. Excellent!

Reviewer's Responses to Questions**Experimental quality**

Does each figure have the proper controls?

If 'No', please indicate reasons in Comments for Author box below.

Reviewer #1:

- Yes

Reviewer #2:

- No

Reviewer #3:

- Yes

Were the data analyzed using appropriate statistical tests?

If 'No', please indicate reasons in Comments for Author box below.

Reviewer #1:

- No

Reviewer #2:

- No

Reviewer #3:

- No

Reproducibility

Were experiments performed using adequate number of biological replicates?

If 'No', please indicate reasons in Comments for Author box below.

Reviewer #1:

- Yes

Reviewer #2:

- Yes

Reviewer #3:

- No

Does the methods section provide sufficient detail to permit reproducibility?
If 'No', please indicate reasons in Comments for Author box below.

Reviewer #1:

- Yes

Reviewer #2:

- Yes

Reviewer #3:

- Yes

Completeness

Are the manuscript's conclusions supported by the data?
If 'No', please indicate reasons in Comments for Author box below.

Reviewer #1:

- Yes

Reviewer #2:

- No

Reviewer #3:

- Yes

Scholarship

Do the authors cite and discuss the merits of data that would argue for and against their conclusion?
If 'No', please indicate reasons in Comments for Author box below.

Reviewer #1:

- Yes

Reviewer #2:

- Yes

Reviewer #3:

- Yes

Does the manuscript title & abstract accurately reflect the contents of the manuscript, without hyperbole?
If 'No', please indicate reasons in Comments for Author box below.

Reviewer #1:

- Yes

Reviewer #2:

- Yes

Reviewer #3:

- Yes

First revision

Author response to reviewers' comments

Reviewer 1

Major comments:

1. In lines 47-57 (page 3), the authors test the potential toxicity of the 3D scaffold by monitoring the presence of SDS using HR-MAS technology. However, this section should also provide details about the scaffold's composition and explain the rationale for checking SDS contamination.

We have slightly reworded this section to make it clearer that the decellularizing agent (SDS) is toxic to worms, and added a citation to the scaffold preparation methods for reference.

2. In Figure 1A and the corresponding results described in lines 21-31 (page 4), no statistical analysis has been performed to support the conclusion regarding differences in pumping rate across rearing conditions. Including appropriate statistical tests is essential to validate this claim.

We assumed here that the reviewer was referring to 2A, not 1A.

We are providing descriptive statistics for all reported data in supplementary material to avoid overcluttering the text.

3. In the section titled “*Developmental timing is altered by experience with the enriched environment*”, the results presented primarily concern morphological traits - body length, body width, area, and perimeter - rather than developmental timing. Did the authors examine whether worms progressed through the developmental stages (L1, L2, L3, L4, and adult) at the same rate or over different durations? If so, those results should be reported here. The current data would be more appropriately placed under a heading such as “*Morphological alterations observed under different rearing conditions.*”

We did not quantify the proportion of larval stages within populations.

We have changed the wording for the heading and explicitly specified this as a next step in the discussion.

4. In Figure 1B, the authors normalized body length by body width, which represent two distinct morphological parameters. The rationale for this choice of normalization is unclear. It would be more informative to present how body length changes over time independently of body width.

We assumed here that the reviewer was referring to 2B, not 1B.

The presented values for body length are normalized to the mean body length of agar:agar worms on day 1; body width is not involved in this figure.

We have added that all values are normalized to their “respective” mean for clarity.

5. In Figures 2 and S2, no statistical analyses are shown. Including appropriate statistical tests is necessary to support the conclusions drawn.

We are providing descriptive statistics for all reported data in supplementary material to avoid overcluttering the text.

6. From Figures 2 and S2, Guiset et al. conclude that scaffold-reared worms are thinner and shorter, with reduced area and perimeter compared to agar-grown worms. Could these differences arise from mechanical compression within the scaffold or from reduced nutrient availability? Did the authors test these possibilities?

Mechanical and food availability variants were not tested, and acknowledged as an important next step in the discussion: “While the scaffold habitat provides structural enrichment, other environmental factors, such as differences in local food availability, oxygen diffusion, or waste accumulation, could be contributing to the observed effects. Controlling for these factors or understanding their dynamics in the scaffold would strengthen the causal links between environmental complexity and phenotypic outcomes.”

“mechanosensory feedback,” was added for completeness.

7. In Figure 2E, it is unclear what each data point represents. Does each point correspond to the Oil Red O staining intensity of an individual worm? If so, under the scaffold:scaffold condition, only two worms appear to have been measured within the 0-2 body length range. The sample size here is very low, and the authors should consider increasing the number of worms analyzed. Additionally, please clarify whether the staining intensity is normalized relative to body size for each worm.

Yes, each data point represents an individual worm's Oil Red O staining intensity, we have clarified this explicitly in the figure legend.

Regarding sample size, we would like to clarify that the total number of worms analyzed per condition is substantial (agar:agar: n=66; agar:scaffold: n=65; scaffold:agar: n=64; scaffold:scaffold: n=57), which we believe is adequate to support the observed trend. Critically, the purpose of this experiment was not to compare fat content at specific developmental stages. This analysis suggests that all groups tend to have increased major fat stores with size, arguing against the possibility of calorie-restriction in the scaffold. We believe that this is well-supported by the current dataset and does not depend on dense sampling within any particular size range. We have clarified this in the results section by changing the wording to “worms of mixed life stages”.

Yes, the staining metric is adjusted by worm size. We have modified the figure's y-axis title to reflect this. This is already described in-text (“We analyzed the staining intensity of individual worms relative to their body size.”) and in the figure legend (“Mean whole body redness by body length stained by Oil Red O.”).

8. In lines 18-20 (page 6), the authors state that “eggs from scaffold-grown worms showed greater elongation, although variability was large across all groups.” However, Figure 3F does not display any data to illustrate this variability. The authors should either include the variability in the figure or clarify how it was assessed.

Figure 3F is presented as a boxplot, which inherently displays variability through the interquartile range (IQR, represented by the box height), the median (horizontal line within the box), whiskers extending to data points within 1.5× IQR, and individual outliers plotted beyond this range. However, we agree that the current wording could be confused with a variability metric and is not strictly necessary since the variability we reference is directly visible, thus this phrase was removed.

9. In lines 5-6 (page 7), the authors state that “only scaffold:scaffold animals were still alive at the 6-hour time point.” Please include the exact percentage of animals that remained alive at this time point for clarity.

Throughout the manuscript, including in this section, the authors describe the observations from their experiments but often do not provide clear inferences or conclusions drawn from these results. For example, what does the increased survival of scaffold:scaffold animals imply about the role of the scaffold environment? Adding interpretive conclusions alongside the reported data would make the findings more meaningful and strengthen the overall impact of the study.

We are providing descriptive statistics for all reported data in supplementary material to avoid overcluttering the text.

Regarding the addition of conclusions in the results section: Review 3 mentions that this was overdone (**R3.3**) and Review 2 mentions that some interpretations were overextended (**R2.4**). We agree that interpretations should be kept to the discussion section, even more so given the exploratory nature of the manuscript. Thus, we have not added any interpretations through the results section, and actually removed most of them (see **R3.3** for list).

10. In lines 4-6 (page 8), the authors state that “we calculated their bend amplitude as a proportion of their body length.” This would be clearer if the authors included a small schematic in Figure 5F to illustrate how bend amplitude was measured.

A schematic was added to the accompanying supplementary figure (Figure S3D).

11. In lines 37-39 (page 8), the authors describe the different shapes adopted by worms (O, U, and 6-shapes). Including representative images of these shapes would be helpful.

We have added the method citation as reference in-text and some representative images as supplemental (Figure S3C). We added this in the swimming section using these shapes, since it comes before the crawling section.

12. For the transcriptional profiling experiments, please provide details on the number of worms used, their developmental stage, and the number of biological replicates included in the RNA sequencing.

Developmental stage: Already in-text and in methods, added to figure legend.

Number of replicates: Added to statistics. Added “biological” in the methods for clarity. Moved within Figure legend for clarity.

Number of worms: Already in methods. We believe that this information is a specific detail that is not required in-text or in the figure legend.

Minor comments:

1. Please ensure that figures are presented in the order in which they are cited in the text. For example:

- Page 4: Figures S2C and S2D appear before S2B.
- Page 6: Figure 4B is cited before 4A.
- Page 7: Figure 5C is cited before 5B.

Thank you for this detail. We have reviewed all figures and adjusted the order where needed. We have not found other instances than the ones listed by the reviewer.

2. Figure 3E is missing from the manuscript text and should be referenced appropriately.

Fixed!

Reviewer 2

The use of decellularized fruit tissue scaffolds introduces multiple potential confounders that are not adequately controlled or quantified in the manuscript. Without these controls, the observed effects cannot be confidently attributed to environmental enrichment per se.

Environmental enrichment (EE) is inherently comparative to standard lab conditions rather than absolute, or attributable to a single variable. EE describes conditions that are more appropriately complex or naturalistic relative to standard laboratory conditions, not the manipulation of a single isolated factor (Renner & Rosenzweig, 1987). The introduction of multiple co-varying factors is not a confound to be eliminated but rather constitutes EE. Paraphrasing Voelkl et al. (2021): The simplicity of our standardized laboratory rearing environments is not neutral: this simplicity is, in itself, an environment that the organism interacts and reacts to and is also multifaceted.

We agree that the scaffold environment introduces multiple variables beyond structural complexity. This is explicitly acknowledged in our limitations section, where we note that "other environmental factors, such as differences in local food availability, oxygen diffusion, or waste accumulation, could be contributing to the observed effects.", and we frame our study precisely as characterizing phenotypic responses to a different environmental context rather than attributing effects to a single causal mechanism.

We have added more details about EE in the introduction in response to R3.2, which we believe adds more context and background about EE literature and will help the reader better understand the approach. We are happy to soften the language or add more details in other sections as well.

The authors have mentioned that the p-values were not computed due to the exploratory nature of the study but even in this case, descriptive statistics accompanied by measures of uncertainty are essential to support interpretation and distinguish true effects from random variation.

We are providing descriptive statistics for all reported data in supplementary material to avoid overcluttering the text.

Although RNA-seq data are presented, the current results are insufficient to connect gene expression changes with observed phenotypes in a mechanistic manner.

We agree that the current data do not establish mechanistic links between transcriptomic changes and phenotypes, nor were we trying to claim otherwise. The goal of this exploratory study is descriptive as environmental enrichment (or simply varying environments) are rarely studied in *C. elegans*. We aimed to show that environmental context does affect *C. elegans* biology across multiple levels (phenotypic, behavioral, and transcriptomic), thereby motivating future mechanistic investigations, but not to explicitly link any genes to any of our measured traits.

We state in the discussion that the RNA-seq analysis "provides a starting point for understanding the molecular mechanisms underlying environmental adaptation." We also explicitly acknowledge in our limitations section that "the specific molecular pathways mediating the observed phenotypic changes require further investigation through targeted genetic manipulations, epigenetic modifications, and the study of inherited small RNAs and proteomic profiles."

To ensure this framing is unambiguous, we have revised the relevant passage in the Discussion.

We hope this clarifies that mechanistic dissection is an explicit goal for future work, not a claim of the present study.

The discussion overextends the findings by implying intergenerational inheritance and environmental adaptation, though the data show only mild, reversible effects which makes the interpretations speculative and hence are not substantiated by the presented results.

We believe that mild, reversible effects do not preclude intergenerational patterns, these represent separate, complementary findings rather than contradictory ones.

The observation that most transcriptional and phenotypic changes reversed within one generation is itself informative: it demonstrates that *C. elegans* exhibits rapid phenotypic

plasticity in response to environmental context. This is expected from its ecology and well-documented in the broader literature. Phenotypic changes operate across multiple timescales: not all environmentally-induced variations are permanent, nor would we expect them to be. Our findings are consistent with normal developmental plasticity rather than evolutionary change. Our observations also suggest that a subset of traits did differ specifically in ancestry-mismatched groups (increased quiescence during swimming, more tortuous crawling paths in the absence of food, altered burrowing success in agar:scaffold worms), which is precisely the pattern one would expect if some intergenerational influence exists alongside the more dominant within-generation plasticity. The identification of Y102A5C.5 and Y102A5C.6 (pseudogenes previously implicated in transgenerational inheritance and learning (Posner et al., 2019; van der Linden et al., 2010)) as ancestry-linked differentially expressed genes provides some molecular clues for this interpretation.

We also wish to note that our study necessarily examined only a subset of the vast array of traits that could be measured in *C. elegans*. We deliberately selected a diverse set of traits spanning life history, behavior, and transcriptomics to broadly characterize environmental responsiveness. That we observed only mild ancestry-dependent effects within this particular selection does not preclude the possibility that other trait choices might reveal stronger or more numerous intergenerational patterns. We believe this is why the interpretations presented in the discussion are relevant. Our observations, considered alongside established work on inheritance in *C. elegans*, suggest that intergenerational effects represent a promising avenue for future investigation. We present these interpretations as well-supported hypotheses to guide such research, not as definitive claims of mechanism.

We also wish to emphasize that our manuscript is explicitly and purposefully framed as exploratory throughout, and we do not claim causal relationships. Our interpretations are grounded in the existing literature and presented as hypotheses to guide future research. Reviewer 3 also encouraged us to provide more interpretations of our findings (R3.4), reflecting that well-supported hypotheses are appropriate and valuable even in exploratory studies. We believe our discussion strikes an appropriate balance between interpreting patterns in light of established literature and acknowledging the preliminary nature of our conclusions. We have still revised phrases in the discussion to ensure internal consistency with our exploratory framing where the language was stronger than warranted.

I would also suggest adding Figure numbers on the figure itself for better quality.

We will gladly add Figure numbers if the Editor thinks this is needed to match the Biology Open styling.

Reviewer 3

Major

Firstly, the authors frame this work throughout as an 'exploratory study'. Given this, they do not provide any real research questions, hypotheses, or statistical tests. Thus, the paper is essentially a presentation of raw data for a wide range of traits that hint at trends but offers little in the way of concrete findings. I still believe this to be a useful contribution and looking at the sample sizes (somewhat hidden in the figure legends) then it seems that although more complex statistical models may not be viable, at least some simple (even if univariate) statistical comparisons may be feasible. The authors claim to be reporting 'effect sizes', but I see nothing clearly denoting this in the paper.

We are providing descriptive statistics for all reported data in supplementary material to avoid overcluttering the text, as well as a compound divergence metric in-text.

Since box plots show data distribution, effect sizes can be visually assessed. However, we have still removed the effect size sentence from the methods section to avoid confusion with actual calculations.

Secondly, and related to the above, the introduction is poorly focussed and brief. I would like to see more engagement with what the paper really aims to do (beyond simply presenting data), and

what the potential applied/translational outcomes of these findings might be. There are only a handful of references in the introduction, showing quite a limited engagement with what is surely a plentiful literature on the broader questions. Given what is known, how might you expect your measured variables to vary across different rearing environments? How might this differ between behaviour, physiology, and genetic factors? I think the broad categorisation used in the discussion is useful for presenting the work clearly - i.e., headings in bold, and you might consider using these throughout aims/methods/results.

We have expanded the introduction with more context from the literature and predictions.

The Biology Open guidelines require that no subheadings be added to the introduction. Our understanding of the guidelines is also that only one level of subheadings is allowed, so we have not added any additional ones in the results and methods.

Thirdly, although well written, the paper is structured in quite an awkward way for the reader. Some of this might be more common in some sub-fields than others, however, I find having a results section which includes both detailed methods and a fair amount of interpretation to be a complex way of presenting the paper. I do understand that there are very many different tests and variables, so a classic methods and results set up might be difficult (as it would be hard to track which methods go with which results). However, I think some careful consideration of the structure is needed to ensure consistency and clarity.

We have modified the results section to include minimal methods explanations (just enough to understand the measurement) while keeping the greater details to the methods section. We have completely removed interpretations from the results section.

We left the one for the crawling body shapes, as it speaks more to what this analysis does than provide an interpretation, and this type of analysis is new.

Finally, much like the discussion, the discussion is a bit vague, and the authors sit on the fence with the data without really engaging with questions or applications. I respect their acceptance of a lack of statistical power - and fully support careful interpretation, but I think more can be said and interpreted and that the authors might take a bit more of a (careful) position in what the paper is about, what the findings are, and (importantly) what the findings might mean. If a bigger study were to hold up your findings, then what would it tell us, and how could this be used?

We have expanded the general interpretations paragraph of the discussion.

Minor

Continuous line numbers are much easier for reviewers, rather than by page.

This was our intention, sorry for the error! This must come from the bioRxiv transfer.

Note: We tried changing our version of the manuscript to new line numbers every page, but none of them seem to match those shared by the reviewer. We did our best to infer which section was referred to in each comment below! We will gladly revisit if necessary.

In general, very clearly written, but I think a much deeper engagement with the literature is required.

We believe this was addressed in **R3.2** by expanding the introduction.

P2; L23-27: this could be clearer and better framed in more specific applied research (e.g., do you mean some importance for biomedical models, behavioural models, etc.)

We inferred this was referring to “Previous research has underscored that minimalistic laboratory conditions can significantly affect gene expression patterns and behaviors, potentially obscuring the full spectrum of an organism’s biological responses (Alfred & Baldwin, 2015; Voelkl et al., 2020).”

Our additions in **R3.2** already expand on this section.

P3; L23-37: here I would like to see some more specific aims. First, I recommend making clear the gap in knowledge, by expanding on the literature used in the intro. For example, what is known in this species, and what about other model species? has anyone ever looked at heritability? How is this work different? Then I think you need to be clear about your contribution and its potential relevance.

We inferred this was referring to the last two paragraphs of the introduction.

We feel like those points have been largely addressed through our additions in **R3.2**, although maybe not to the level that the review would have liked. However, we feel comfortable about this level of detail, as we do mention that “implications of raising *C. elegans* in alternative environments remain largely unexplored.”, meaning that there is really no other comparable literature in *C. elegans* at the moment.

P3; L23-37: the aims need to clearly set up the general variables used, e.g., refer to behaviour, life history, heritability, genetics...otherwise the methods seem to come out of nowhere.

We inferred this was referring to the last two paragraphs of the introduction.

Our additions in **R3.2** already expand on this.

This is more of a decision for the Editor/journal, but I found all the methods and discussion embedded into the results section confusing, as per my previous comment. Thus, I think the structure could be improved. I have not marked all the cases of what are clearly methods/discussion throughout, as it is pervasive - so if you/editor agree with any restructure then please check carefully.

We have addressed this in **R3.3**.

P3; L47: 'the' potential toxicity?

We inferred this was referring to “To ensure that phenotypic effects were not confounded with potential toxicity of the [...]”

This was fixed in **R1.1**.

P3; L49: what is toxic about the preparation process?

We inferred this was referring to “To ensure that phenotypic effects were not confounded with potential toxicity of the [...]”

This was fixed in **R1.1**.

P4; L27-31: this is discussion - try and keep any interpretation for the discussion section.

We inferred this was referring to “This range is consistent with typical pumping rates [...]”

This was fixed in **R3.3**.

P5: is habitat experience the same thing as environmental context? Clarify and if so ensure consistency.

We inferred this was referring to “Fat content is not affected by habitat experience”

Fixed! We did not find other mentions of the “habitat experience” term.

P5; L25-27: this is discussion. In most fields, citations are avoided in results.

We inferred this was referring to “Importantly, no condition showed [...].”

This was fixed in **R3.3**.

P5; L49-51: this is discussion.

We inferred this was referring to “This suggests a potential intergenerational [...].”

This was fixed in **R3.3**.

P7; L36: 'ancestors particularly had' reword?

“Particularly” was removed.

(Discussion) Generally good, and better framed in the literature. However, please see major comment. I like the framing around broad categories for the variables, and as per previous comments I think this could be expanded throughout.

We have expanded part of the discussion in **R3.4**.

While we would also like to discuss our results more lengthily, we are limited by the maximum word count. We also feel like most of the discussion is sufficiently detailed - scoped “as a whole” instead of for each trait individually which matches the general exploratory nature of the paper.

P11; L11-29: this section needs references to published literature.

We inferred this was referring to “Transcriptomic profiles of environmental adaptation.” paragraph.

We expanded this paragraph with parallels to the literature.

P12; L52-58: As per my previous comment, this final section needs to refer back to what your aims are, make some conclusions about what your findings mean (in more concrete terms) and make some effort to say what their relevance might be for future study or application.

We inferred this was referring to the very last paragraph of the discussion “In conclusion, [...].”

We feel like the last paragraph represents a broader concluding statement and significance. We have addressed the reviewer’s comment about aims/conclusions when addressing comment **R3.4**.

(Methods) Are very well written and clear. Perhaps they could be organised around the broad headings previously mentioned?

Same as **R3.2**. Our understanding is that only 1 level of subheadings is allowed. The methods are already presented in the same order in the results, so it should be relatively easy to follow for the readers.

P18; L27- here I think some more effort could be made to test some hypotheses. I understand the limitations of sample size, but I think you either need to accept that more work is needed to increase these, or have a go at interpreting what you can with the limited data.

We inferred this was referring to the “Statistical analysis” section.

We are providing descriptive statistics for all reported data in supplementary material to avoid overcluttering the text, as well as a compound divergence metric in-text.

P18; L42-44: what does 'focussing on estimating effect sizes' mean, and is it part of a statistical analysis? I don't see anything reported on this, e.g., odds ratios, Cohen's d, etc.

We inferred this was referring to the second paragraph of the “Statistical analysis” section.

This was addressed in **R3.1**.

P18; L46: I don't see much value in having an aim to generate hypotheses in published work, and I don't see that you follow this through. Similarly, do you 'uncover novel insights without the constraints of predefined hypotheses? I am not sure I agree with the general approach. You must have had some thoughts about how the environment would impact on the variables?

We inferred this was referring to the “Statistical analysis” section.

We agree that our framing overstated the hypothesis-free nature of this work. We did enter this study with the expectation that environmental enrichment would affect *C. elegans* biology, but with no hypothesis on the individual traits given the lack of prior enrichment studies in *C. elegans*. We measured many traits without predictions about which would be affected or to what degree, and we deliberately avoided formal hypothesis testing to prevent inflating false positives across multiple comparisons, a situation well described here (<https://elifesciences.org/articles/48175>) and that can be visualized here (<https://www.react-graph-gallery.com/example/t-test-playground>).

This is also a well-established concern in the philosophy of science (Popper's falsificationism, the replication crisis, etc.) that hypothesis-driven research can inadvertently introduce confirmatory bias, where investigators (consciously or not) "fish" for results that support their predictions. By measuring a broad suite of traits with equal rigor regardless of our expectations, we are at even greater risk of this error. We aimed to let the data guide our conclusions/future research rather than selectively emphasizing results that confirmed preconceptions. We believe this approach aligns with ideals of objectivity in scientific inquiry during exploratory work, even if our original wording suggested a stronger claim than intended.

We have reworded the second paragraph of this section to reflect this better, and we hope that the addition of the compound metric also better reflects our approach/perspective/hypothesis.

Might the sample sizes be better reported in the methods section? (or repeated)

It is not customary in *C. elegans* research to report sample sizes in the methods section.

<https://journals.biologists.com/jeb/article/211/23/3703/17953/Episodic-swimming-behavior-in-the-nematode-C>

<https://www.pnas.org/doi/full/10.1073/pnas.1909210116#sec-3>

Second decision letter

MS ID#: bio.062282R1

MS Title: The impact of rearing environment on *C. elegans*: Phenotypic, transcriptomic and intergenerational responses to 3D enriched habitats

Authors: Aurélie Guisnet, Nour Halaby; Maxime Rivest; Beatriz Romero Quineche; Michael Hendricks

I am happy to tell you that your manuscript has been accepted for publication in Biology Open, pending our standard publication integrity checks. It was accepted on 26th January 2026.